# SHARPNESS-AWARE MINIMIZATION CAN HALLUCINATE MINIMIZERS

## ABSTRACT

Sharpness-Aware Minimization (SAM) is a widely used method that steers training toward flatter minimizers, which typically generalize better. In this work, however, we show that SAM can converge to *hallucinated minimizers*—points that are not minimizers of the original objective. We theoretically prove the existence of such hallucinated minimizers and establish conditions for local convergence to them. We further provide empirical evidence demonstrating that SAM can indeed converge to these points in practice. Finally, we present a simple yet effective remedy for avoiding hallucinated minimizers.

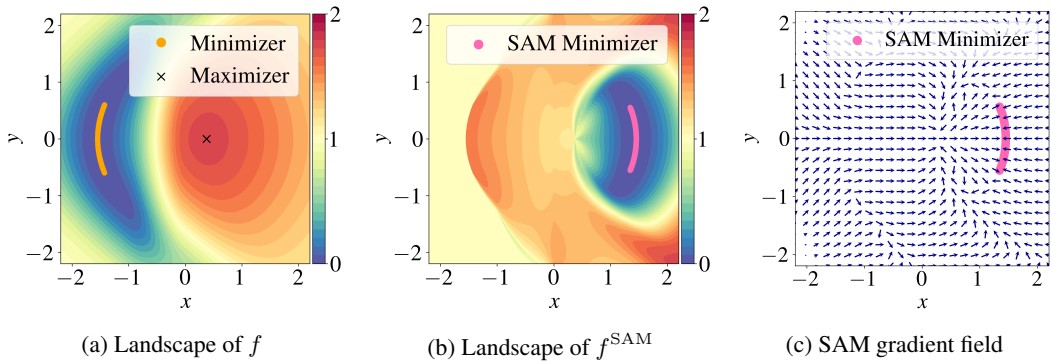

(a) Landscape of $f$      (b) Landscape of $f^{\text{SAM}}$      (c) SAM gradient field

Figure 1: Illustrative example of hallucinated minimizers. See Appendix D for details. (a) Smooth function $f$ with a minimizer set and an isolated maximizer. (b) $f^{\text{SAM}} = f(x + \rho \frac{\nabla f(x)}{\|\nabla f(x)\|})$; its minimizers do not correspond to minimizers or stationary points of $f$ and are therefore hallucinated. (c) Vector field of $\nabla f(x + \rho \frac{\nabla f(x)}{\|\nabla f(x)\|})$; the hallucinated minimizers are attractors of the SAM iteration.

## 1 INTRODUCTION

It has been empirically observed in deep learning that flat minimizers tend to generalize better than sharp ones (Neyshabur et al., 2017; Jiang et al., 2020). Motivated by this observation, Sharpness-Aware Minimization (SAM) was proposed as a training method that fits the data while simultaneously regularizing against the sharpness (Foret et al., 2021; Zheng et al., 2021; Wu et al., 2020). Specifically, for a differentiable function $f\colon \mathbb{R}^d \to \mathbb{R}$, SAM minimizes the objective

$$f^{\text{SAM}}(x) := f\left(x + \rho \frac{\nabla f(x)}{\|\nabla f(x)\|}\right),$$

where the perturbation radius $\rho > 0$ controls the strength of the regularization. By encouraging convergence to flat minimizers, SAM has demonstrated strong empirical performance and has inspired a variety of practical variants (Foret et al., 2021).

However, although prior theoretical studies have analyzed the convergence of SAM to a minimizer under various conditions, most rely on assumptions that rarely hold in deep learning, such as convexity (Si & Yun, 2024) or a decaying perturbation radius (Khanh et al., 2024; Oikonomou & Loizou, 2025). Consequently, the validity of these convergence results is not guaranteed in practical settings.

**Contributions.** In this paper, we theoretically and empirically demonstrate that SAM can, in fact, *hallucinate* minimizers, in the sense that its iterates may converge to points that are not minimizers of the original objective, as illustrated in Figure 1. This finding reveals a previously unrecognized failure mode of SAM in deep learning, one that is fundamentally distinct from issues caused by local minima or saddle points. To address this, we present a simple switching strategy that effectively avoids hallucinated minimizers.

## 1.1 RELATED WORK

**SAM and its variants.** Building on the observation that flat minimizers are stable under small perturbations (Hochreiter & Schmidhuber, 1997; Keskar et al., 2017; Chaudhari et al., 2019), SAM was proposed as a method to seek such minimizers (Foret et al., 2021). SAM has demonstrated remarkable performance across a wide range of deep learning tasks (Foret et al., 2021; Bahri et al., 2022; Zhong et al., 2022; Lee et al., 2023), motivating numerous extensions.

One line of work improves SAM by modifying its perturbation direction. Kwon et al. (2021) adapted the perturbation in a scale-invariant manner, while Kim et al. (2022) redefined it using Fisher information geometry. Li et al. (2024) removed the full-batch gradient from the perturbation direction and leveraged stochastic gradient noise for generalization. Li & Giannakis (2024) incorporated momentum into the perturbation to suppress variance and stabilize the adversary. Instead of altering the perturbation itself, Zhuang et al. (2022) adjusted the gradient update through orthogonal decomposition to reduce the surrogate gap.

Another line of research addresses the computational overhead of SAM, which stems from requiring two gradient computations per step. Some approaches reduce the number of gradient evaluations: Liu et al. (2022) computed the perturbation only periodically, Jiang et al. (2023) activated SAM only when the gradient norm is large, and Du et al. (2022b) reused gradients to avoid the second computation. Others restrict the scope of perturbations: Du et al. (2022a) applied them only to a random subset of parameters and sharpness-sensitive data, while Mueller et al. (2023) showed that limiting them to normalization layers preserves most of the benefits. Beyond reducing computation, Xie et al. (2024) improved training efficiency by parallelizing the two gradient computations.

**Theoretical analyses of SAM.** Alongside its practical success, a growing body of theoretical work has analyzed SAM and examined its generalization properties from multiple perspectives. Wen et al. (2022) formalized the precise notion of sharpness minimized by SAM, clarifying its regularization effect. Möllenhoff & Khan (2023) reinterpreted SAM as a relaxation of Bayesian inference. Chen et al. (2023) showed that SAM mitigates noise fitting and improves generalization over stochastic gradient descent. Wei et al. (2023) further demonstrated that SAM alone can enhance adversarial robustness while maintaining clean accuracy.

Another line of work investigates the training dynamics and stability of SAM. Compagnoni et al. (2023) analyzed SAM through the lens of stochastic differential equations, offering a continuous-time perspective. Bartlett et al. (2023) studied quadratic objectives, showing how SAM oscillates across narrow valleys before drifting toward wider minimizers. Dai et al. (2023) demonstrated that normalization plays a key role in stabilizing SAM and ensuring robustness. Long & Bartlett (2024) extended the edge-of-stability threshold of gradient descent to SAM, showing that it depends on the gradient norm. More recently, Zhou et al. (2025) highlighted a late-phase effect, whereby SAM selects flatter minimizers when applied in the later stages of training.

Finally, several works have investigated the convergence properties of SAM in diverse settings. Andriushchenko & Flammarion (2022) proposed USAM, an unnormalized variant obtained by removing gradient normalization, and analyzed its convergence. Si & Yun (2024) provided a systematic study across convex, strongly convex, and nonconvex regimes. Khanh et al. (2024) developed a convergence analysis of SAM and its variants within the framework of inexact gradient descent. Most recently, Oikonomou & Loizou (2025) analyzed SAM and USAM within a unified framework and proved convergence under the Polyak–Łojasiewicz condition.

**Hallucinated minimizers and our contribution.** Several prior studies have reported that SAM may converge to points that are not minimizers of the original loss. Kaddour et al. (2022) empirically observed that SAM can become trapped at saddle points, and Kim et al. (2023); Compagnoni et al.

(2023) provided a theoretical explanation of this phenomenon using a continuous-time model of SAM. In another line of work, Bartlett et al. (2023) showed that the SAM update is equivalent to gradient descent on a surrogate function in the quadratic case, and Si & Yun (2024) proposed a virtual loss to extend this idea, although it is rigorously defined only in one dimension and lacks guarantees in higher dimensions.

In contrast, hallucinated minimizers represent a fundamentally different failure mode of SAM. They differ from saddle points in that they are not critical points of the original loss, and from surrogate-based interpretations in that they arise directly from the SAM objective. Our analysis applies to general nonconvex and high-dimensional settings and provides a rigorous characterization of hallucinated minimizers that can emerge in practical deep learning scenarios.

## 1.2 PRELIMINARIES AND NOTATION

Throughout the paper, we denote $u(x) := \nabla f(x)/\|\nabla f(x)\|$ (for $x$ such that $\nabla f(x) \neq 0$). To optimize the SAM objective $f^{\mathrm{SAM}}(x)$, we require its gradient $\nabla f^{\mathrm{SAM}}(x)$. Under the standard smoothness assumption on $f$, this gradient is given by

$$\nabla f^{\mathrm{SAM}}(x) = \big(I + \rho\,\nabla u(x)\big)\,\nabla f\big(x + \rho\,u(x)\big),$$

where $\nabla u(x)$ denotes the Jacobian of $u(x)$. In practice, however, for computational simplicity, one does not use the exact gradient $\nabla f^{\mathrm{SAM}}$. Instead, SAM employs the "shifted" gradient $\nabla f\big(x + \rho\,u(x)\big)$. This yields the (full-batch) SAM iteration

$$x_{k+1} = x_k - \eta_k\,\nabla f\big(x_k + \rho\,u(x_k)\big), \qquad k = 0, 1, 2, \dots,$$

where $x_0 \in \mathbb{R}^d$ is the starting point and $\eta_0, \eta_1, \dots \in \mathbb{R}_+$ is the sequence of step sizes. The perturbation radius $\rho > 0$ controls the degree of "flatness": larger values of $\rho$ expand the neighborhood over which the loss is minimized, thereby steering the SAM iteration towards flatter minimizers. In practice, SAM is typically implemented with stochastic gradients. When clarification is needed, we refer to *SAM with full-batch gradients* versus *SAM with stochastic gradients*.

We introduce some notation. A function $f\colon \mathbb{R}^d \to \mathbb{R}$ is called *real-analytic* if its Taylor series at any point $x_0$ converges to $f$ on a neighborhood of $x_0$. For $\alpha \in \mathbb{R}$, the $\alpha$-superlevel set of $f$ is defined as $\{x\colon f(x) \geq \alpha\}$. For a set $C \subset \mathbb{R}^d$, we write $\partial C$ for its boundary. A set $C$ is *connected* if it cannot be expressed as the union of two disjoint, nonempty open sets. The distance from a point $x \in \mathbb{R}^d$ to a nonempty set $C \subseteq \mathbb{R}^d$ is $d(x, C) := \inf_{y \in C} \|x - y\|$; if $C$ is closed, the infimum is attained, and hence $d(x, C) = \min_{y \in C} \|x - y\|$. For $\delta > 0$, $B_\delta(x)$ is the ball centered at $x$ with radius $\delta$ and the (closed) $\delta$-neighborhood of $C$ is $\mathcal{N}_\delta(C) = \{x\colon d(x, C) \leq \delta\}$.

## 2 EXISTENCE OF HALLUCINATED MINIMIZERS

In this section, we establish—under very mild assumptions—the existence of hallucinated minimizers: local minimizers of $f^{\mathrm{SAM}}$ that are not even stationary points of the original function $f$. Formally, we define hallucinated minimizers as follows:

**Definition.** A point $x \in \mathbb{R}^d$ is a *hallucinated minimizer* of $f\colon \mathbb{R}^d \to \mathbb{R}$ for $\rho > 0$ if $x$ is a local minimizer of $f^{\mathrm{SAM}} = f\left(x + \rho \frac{\nabla f(x)}{\|\nabla f(x)\|}\right)$ while satisfying $\nabla f(x) \neq 0$.

When the loss function $f$ is convex, hallucinated minimizers cannot arise, as shown in Theorem A.1 of Appendix A. In practical deep learning, however, the loss function $f$ is highly nonconvex, and local *maximizers* of $f$ can give rise to hallucinated minimizers.

## 2.1 SIMPLIFIED EXISTENCE PROOF WITH ISOLATED MAXIMIZERS

We begin with a simplified proof under the more restrictive assumption that $f$ has an isolated local maximizer, defined as a point $x^\bullet$ with an open neighborhood $U$ such that

$$\nabla f(x) \neq 0 \;\; \text{and} \;\; f(x) < f(x^\bullet) \;\;\; \text{for all } x \in U \setminus \{x^\bullet\}.$$

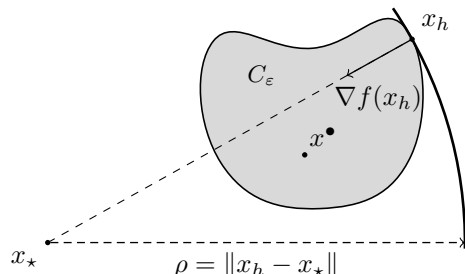

$x_\star$: global minimizer
$x^\bullet$: local maximizer
$C_\varepsilon$: superlevel set near $x^\bullet$
$x_h$: farthest from $x_\star$ on $C_\varepsilon$
$\nabla f(x_h)$ points toward $x_\star$

Figure 2: Illustration of the proof for Theorem 2.1. The point $x_h$ is the farthest from $x_\star$ among the points in $C_\varepsilon$. By the method of Lagrange multipliers, its gradient $\nabla f(x_h)$ points exactly toward $x_\star$.

**Theorem 2.1.** *Let $f : \mathbb{R}^d \to \mathbb{R}$ be continuously differentiable. Assume $f$ has a global minimizer (not necessarily unique) and an isolated local maximizer. Then, a hallucinated minimizer exists for some $\rho > 0$.*

*Sketch of proof.* We provide a brief sketch of the argument, with full details deferred to Appendix A.1. Figure 2 illustrates the key idea of the construction.

Let $\varepsilon > 0$ and define $C_\varepsilon$ as the $(f(x^\bullet) - \varepsilon)$-superlevel set restricted to a neighborhood of the isolated local maximizer $x^\bullet$. For sufficiently small $\varepsilon > 0$, the set $C_\varepsilon$ is compact and satisfies: $(i)$ for every $x \in C_\varepsilon$, we have $f(x^\bullet) - \varepsilon \le f(x) \le f(x^\bullet)$; $(ii)$ the gradient $\nabla f$ does not vanish on $C_\varepsilon \setminus \{x^\bullet\}$; and $(iii)$ $f(x) = f(x^\bullet) - \varepsilon$ on the boundary $\partial C_\varepsilon$.

Next, consider $g(x) = \|x - x_\star\|^2$ and define

$$x_h \in \arg\max_{x \in C_\varepsilon} g(x) = \arg\max_{x \in \partial C_\varepsilon} g(x).$$

In words, if $x_h$ maximizes $g$ over $C_\varepsilon$, then it must lie on the boundary $\partial C_\varepsilon$. Since $\partial C_\varepsilon$ coincides with $\{x : f(x) = f(x^\bullet) - \varepsilon\}$ near $x_h$, we apply the method of Lagrange multipliers to $x_h \in \arg\max_{x \in \partial C_\varepsilon} g(x)$ to obtain

$$2(x_\star - x_h) = \nabla g(x_h) = \lambda \nabla f(x_h).$$

The fact that $\lambda > 0$ follows from the observation that $\nabla f(x_h)$ points "toward" both $x_\star$ and $x^\bullet$, as illustrated in Figure 2. Finally, setting $\rho = \lambda \|\nabla f(x_h)\|/2$ yields

$$x_h + \rho \frac{\nabla f(x_h)}{\|\nabla f(x_h)\|} = x_\star,$$

and hence $f^{\mathrm{SAM}}(x_h) = f(x_\star)$. Thus, $x_h$ is a (global) minimizer of $f^{\mathrm{SAM}}$. $\qquad\square$

Importantly, the existence of a hallucinated minimizer also holds when $x_\star$ is a *local* minimizer, provided that $f$ has a locally Lipschitz gradient. See Appendix A.2 for details.

The proof of Theorem 2.1 reveals the conditions under which a hallucinated minimizer is likely to arise. Figure 2 illustrates the core idea: given a minimizer $x_\star$, $x_h$ satisfies $x_\star = x_h + \rho \frac{\nabla f(x_h)}{\|\nabla f(x_h)\|}$. This means that at $x_h$, the gradient points directly toward $x_\star$. However, near a minimizer, the gradient typically points outward, making it difficult to identify such $x_h$ in its immediate neighborhood. To resolve this, Theorem 2.1 requires the presence of a nearby maximizer, which allows the gradient to align in the desired direction and thus necessitates nonconvexity of the objective.

The proof also shows that the perturbation radius $\rho$ must exceed the distance between minimizer and the maximizer, since $x_h$ is the farthest point from $x_\star$ on the superlevel set $C_\varepsilon$. Because $x_h \in C_\varepsilon$, a hallucinated minimizer is located near the maximizer. This implies that hallucinated minimizers are typically associated with high loss values.

Taken together, these observations suggest that hallucinated minimizers generally arise in nonconvex objectives containing both local maximizers and minimizers, and that they tend to occur near a local maximizer within a high-loss region.

## 2.2 EXISTENCE OF HALLUCINATED MINIMIZERS FOR NEURAL NETWORKS

Theorem 2.1 assumes that the local maximizer is *isolated*. We now relax this assumption, since in neural networks, maximizers (like minimizers) often occur as sets.

To this end, we leverage the real-analyticity of neural networks, an assumption that holds when training on a finite dataset (empirical loss) with real-analytic activation functions. We note that all commonly used activation functions are real-analytic, except for ReLU. Technically, real-analyticity affords us the Łojasiewicz inequality, which we use to rule out certain pathological cases.

Notably, our result does not rely on restrictive but common structural assumptions on the loss function, such as global smoothness or a quadratic form.

**Definition.** Let $f\colon \mathbb{R}^d \to \mathbb{R}$ be a continuous function. A nonempty connected set $X^\bullet \subseteq \mathbb{R}^d$ is a *local maximizer set* of $f$ if there exists $\delta > 0$ such that $X^\bullet$ is the maximizer set over its $\delta$-neighborhood. In other words,

$$X^\bullet = \arg\max_{y \in \mathcal{N}_\delta(X^\bullet)} f(y).$$

Furthermore, we denote by $f(X^\bullet)$ the common function value of $f$ on $X^\bullet$.

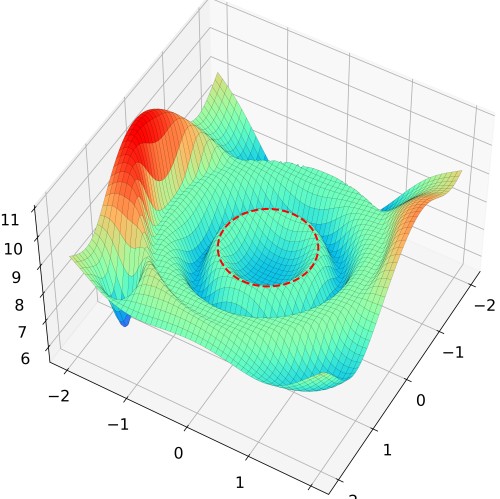

Figure 3: Illustration of a local maximizer set. The dotted unit circle is a local maximizer set of a real-analytic function and is not a singleton.

**Theorem 2.2.** *Let $f\colon \mathbb{R}^d \to \mathbb{R}$ be real-analytic. Assume $f$ has a global minimizer (not necessarily unique) and a bounded local maximizer set. Then, a hallucinated minimizer exists for some $\rho > 0$.*

The proof of Theorem 2.2, fully presented in Appendix A.4, is analogous to that of Theorem 2.1, except that the isolatedness assumption is replaced by the real-analyticity assumption. A key technical point is to rule out the possibility that critical points accumulate densely around $X^\bullet$. This is achieved using the Łojasiewicz inequality:

**Lemma 2.3** (Łojasiewicz (1963)). *If $f\colon \mathbb{R}^d \to \mathbb{R}$ is real-analytic, then for every $p \in \mathbb{R}^d$, there exist an open neighborhood $U$ of $p$, a constant $C > 0$, and an exponent $q \in (0,1)$ such that*

$$|f(p) - f(x)|^q \le C\|\nabla f(x)\| \quad \text{for all } x \in U.$$

**Most modern neural networks are real-analytic.** The real-analyticity assumption on $f$ is practical in the context of deep learning. In particular, consider the empirical loss function

$$f(\theta) = \sum_{i=1}^{N} \ell(h_\theta(X_i), Y_i),$$

where $\theta \in \mathbb{R}^d$ denotes the neural network parameters. If the dataset is finite ($N < \infty$), $\ell(\cdot, \cdot)$ is real-analytic in its first argument (as is the case for most commonly used losses, such as cross-entropy), and $h_\theta$ is a neural network built from real-analytic activation functions (e.g., tanh, ELU, GELU, SiLU, swish), then $f(\theta)$ is real-analytic. More concretely, $h_\theta$ may be a finite composition of linear layers, convolution, attention, residual connections, layer normalization, batch normalization, real-analytic activation functions, the softmax function, average pooling, and dropout. However, $h_\theta$ cannot incorporate ReLU, leaky ReLU, or max-pooling, since these operations are non-smooth and therefore non-analytic.

Further discussion and details are provided in Appendix A.3.

## 3 GEOMETRIC AND DYNAMICAL PROPERTIES OF HALLUCINATED MINIMIZERS

In this section, we establish a finer geometric property of hallucinated minimizers as well as a dynamical property of the SAM iterates. Recall that we use the notation $u(x) = \nabla f(x)/\|\nabla f(x)\|$.

### 3.1 THE SET OF HALLUCINATED MINIMIZERS CAN HAVE MANIFOLD STRUCTURES

The following theorem establishes that when the set of true minimizers has an $m$-dimensional manifold structure, the set of hallucinated minimizers inherits the same geometric structure. In particular, this result explains why, in Figure 1, the set of SAM minimizers appears as a curve.

**Theorem 3.1.** *Suppose $f : \mathbb{R}^d \to \mathbb{R}$ satisfies the assumptions of Theorem 2.2. Assume $\mathcal{M} \subseteq$ $\operatorname{argmin} f$, where $\mathcal{M} \subseteq \mathbb{R}^d$ is a nonempty smooth $m$-dimensional manifold. Let $x_h$ be a hallucinated minimizer with a corresponding $\rho > 0$ as constructed in the proof of Theorem 2.2. If $I + \rho \nabla u(x_h) \in \mathbb{R}^{d \times d}$ is nonsingular, then the set of hallucinated minimizers contains a smooth manifold of dimension $m$.*

We defer the proof to Appendix B.1.

### 3.2 HALLUCINATED MINIMIZERS ARE ATTRACTORS

We have established the existence of hallucinated minimizers, but does SAM actually converge to them? Recall that the SAM iteration is given by

$$x_{k+1} = x_k - \eta_k \nabla f(x_k + \rho \, u(x_k)), \qquad k = 0, 1, 2, \ldots,$$

where $\eta_k > 0$ denotes the step size.

The answer is yes—hallucinated minimizers can indeed be attractors of the SAM dynamics. Thus, the concern about hallucinated minimizers in neural network training is not merely hypothetical. In Section 4, we provide empirical evidence that SAM can converge to hallucinated minimizers. In this subsection, we theoretically establish local convergence to hallucinated minimizers.

**Theorem 3.2.** *Suppose $f : \mathbb{R}^d \to \mathbb{R}$ is real-analytic, and let $H \subset \mathbb{R}^d$ be a bounded, connected set of hallucinated minimizers of $f$ for a fixed perturbation radius $\rho > 0$. Assume there exists $\delta > 0$ such that the $\delta$-neighborhood of $H$ contains no minimizers of $f^{\text{SAM}}$ other than those already in $H$. Assume further that every $x_h \in H$ satisfies*

$$1 + \rho \lambda_{\min}(\operatorname{Sym}(\nabla u(x_h))) > 0, \quad \text{where } \operatorname{Sym}(\nabla u(x_h)) = \frac{1}{2}(\nabla u(x_h) + \nabla u(x_h)^\top).$$

*If the initialization $x_0$ is chosen sufficiently close to $H$, then there exists a sufficiently small fixed step size $\eta_k = \eta > 0$ such that the SAM iterates converge to $H$, in the sense that $d(x_k, H) \to 0$.*

We defer the proof to Appendix B.2.

## 4 EMPIRICAL ANALYSES IN DEEP LEARNING

In this section, we empirically validate our theory by analyzing hallucinated minimizers in deep learning. We show that SAM trajectories can, in practice, converge to hallucinated minimizers. We further demonstrate that a simple switching strategy can effectively prevent this convergence, providing a practical safeguard for SAM against this failure mode.

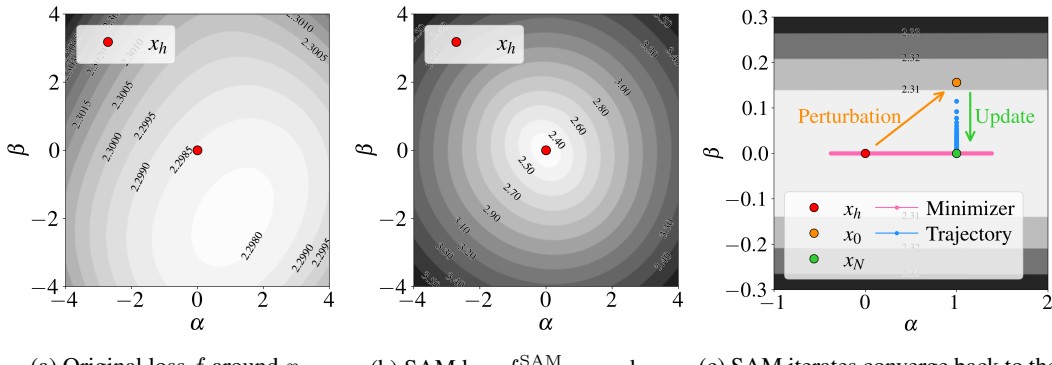

(a) Original loss $f$ around $x_h$

(b) SAM loss $f^{\text{SAM}}$ around $x_h$

(c) SAM iterates converge back to the hallucinated minimizer set.

Figure 4: Visualizations of $f$ and $f^{\text{SAM}}$ around the hallucinated minimizer $x_h$. Plots (a) and (b) are taken on a 2-dimensional plane defined by $x_h$ and two random directions. These show that $x_h$ is indeed a minimizer of $f^{\text{SAM}}$ but not a stationary point of $f$. Plot (c) depicts $f^{\text{SAM}}$ on the 2-dimensional plane containing $x_h$, $x_0$, and $x_N$, where $x_0$ is a small perturbation of $x_h$ and $x_N$ is obtained after $N = 1000$ SAM steps from $x_0$. The pink horizontal line segment indicates the set of hallucinated minimizers, showing that the SAM trajectory converges back to this set.

### 4.1 SAM CAN CONVERGE TO HALLUCINATED MINIMIZERS

We first present an example in which SAM converges to a non-minimizer and confirm that the point is a hallucinated minimizer. We then examine how frequently such convergence occurs across diverse experimental settings.

**Do we really converge to hallucinated minimizers?** We begin with a simple neural network setting and provide an example where SAM converges to a hallucinated minimizer. Specifically, we train a two-layer neural network with `Tanh` activations on MNIST (LeCun et al., 1998) using full-batch updates, yielding a smooth objective. The experimental setup is described in Appendix C. With a perturbation radius $\rho = 1.8$, we train for over 20 million steps and observe that the trajectory converges to a single point $x_h$. At this point, the SAM gradient nearly vanishes ($\|\nabla f(x_h + \rho\, u(x_h))\| = 4.8 \times 10^{-9}$), while the original gradient remains relatively large ($\|\nabla f(x_h)\| = 0.0627$). Thus, $x_h$ is a stationary point of $f^{\text{SAM}}$ but not of $f$, showing that SAM can converge to a hallucinated stationary point.

To verify that $x_h$ is indeed a *hallucinated minimizer*, we visualize the loss landscape around it, as shown in Figure 4. Following the method of Li et al. (2018), we define the plane
$$x(\alpha, \beta) = x_h + \alpha u + \beta v,$$
where $u, v \in \mathbb{R}^d$ are orthogonal vectors of equal norm. In Figure 4a, $x_h$ is clearly not a minimizer of $f$, whereas in Figure 4b, on the same plane, it appears as a minimizer of $f^{\text{SAM}}$. This demonstrates that SAM can converge to a hallucinated minimizer in neural networks.

To investigate the geometry around $x_h$ in more detail, we add a small random perturbation to $x_h$, yielding a nearby point $x_0$ with $\|x_0 - x_h\| = 0.1$. Starting from $x_0$, we run $N = 1000$ additional SAM steps to reach $x_N$. Figure 4c shows the SAM objective on the plane spanned by $x_h$, $x_0$, $x_N$. The SAM trajectory (blue) is projected onto this plane, while the SAM minimizers (pink) are computed within a tolerance of $10^{-9}$. We observe that hallucinated minimizers form a connected set, consistent with prior work on connected minimizers in neural networks (Garipov et al., 2018) and with Theorem 3.1, which guarantees that this structure is preserved.

**How common are hallucinated minimizers?** Next, we investigate convergence to hallucinated minimizers across multiple experiments, showing that this phenomenon can occur frequently in deep learning. Figure 5 presents SAM training outcomes in the same full-batch setting as before, with 80 distinct seeds and 100,000 iterations for perturbation radii $\rho = 1.0, 1.3, 1.6, 1.9$. The top row shows SAM-only results, while the bottom row corresponds to the switching strategy discussed later. In this subsection, however, we focus on the SAM-only setting.

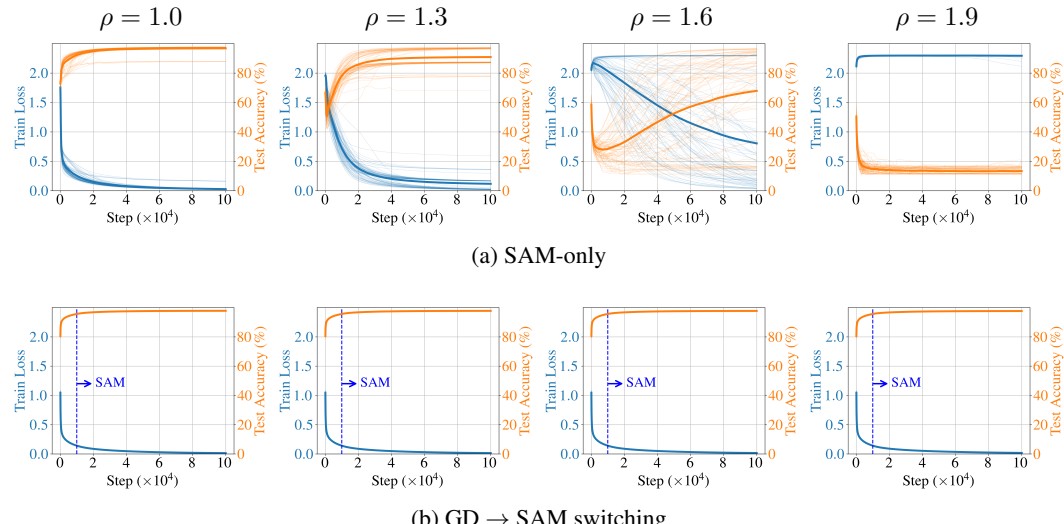

(a) SAM-only

(b) GD → SAM switching

Figure 5: Training loss and test accuracy for SAM on MNIST with a 2-layer network using full-batch gradients. The top row shows the results from SAM-only training, while the bottom row shows the results from the switching strategy as described in Section 4.2. Final average test accuracies are 96.77%, 90.85%, 67.88%, and 13.36% for SAM-only training, compared to 97.77%, 97.73%, 97.69%, and 97.62% for the switching strategy. These results demonstrate that the switching strategy improves test accuracy and stabilizes outcomes across perturbation radii.

The experiments reveal that convergence to hallucinated minimizers depends on the perturbation radius $\rho$. At $\rho = 1.0$, training consistently converges to zero loss, indicating convergence to a true minimizer of the original loss. At $\rho = 1.3$ and $1.6$, some trajectories stabilize at nonzero-loss points. At $\rho = 1.9$, most trajectories converge to such nonzero-loss points, indicating that SAM predominantly reaches hallucinated minimizers.

These findings provide strong evidence that hallucinated minimizers are not rare anomalies but occur consistently in deep learning when the perturbation radius $\rho$ is large. This observation aligns with Theorem 2.1, whose proof requires the perturbation radius $\rho$ to exceed the distance between a minimizer and a maximizer. At the same time, even for the same $\rho$, trajectories may converge either to a hallucinated minimizer or to a true minimizer, consistent with Theorem 3.2, which depends on the initialization's proximity to a hallucinated minimizer.

In the stochastic case, we train ResNet-18 (He et al., 2016) on CIFAR-100 (Krizhevsky, 2009) with mini-batch updates, following the FSAM implementation of Li et al. (2024). We observe a similar trend: larger perturbation radii lead to unstable training. Further details of the experimental settings and results are provided in Appendix C.

## 4.2 SWITCHING STRATEGY FOR AVOIDING HALLUCINATED MINIMIZERS

One obvious approach to avoiding hallucinated minimizers is to use a small perturbation radius $\rho > 0$. Indeed, hallucinated minimizers do not arise when $\rho = 0$ (i.e., when SAM is not applied). However, the perturbation itself is the key mechanism that regularizes against sharpness, making it desirable to use a moderately large value of $\rho$.

In this subsection, we introduce a simple yet effective heuristic for avoiding hallucinated minimizers, which we call *switching*. The idea is to use plain gradient descent for the first 10% of training iterations and then switch to SAM.

As shown in Figure 5b, the switching strategy consistently drives the training loss to zero across all tested perturbation radii, including large values of $\rho$ for which standard SAM fails to converge to the true minima. Figure 6 further demonstrates the improved *test* accuracy achieved under the switching strategy. Not only does switching yield higher test accuracy, but it also reduces sensitivity to the

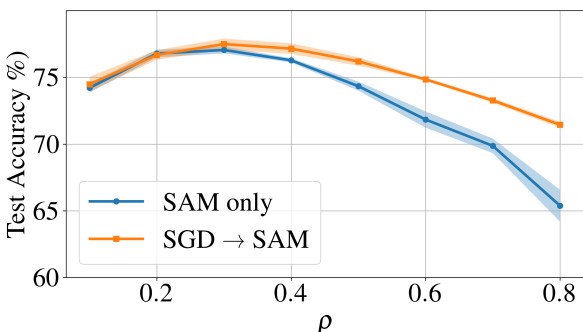

Figure 6: Final test accuracy for SAM-only and the switching strategy on CIFAR-100 with ResNet-18 using stochastic gradients. Each curve shows the mean (bold) and standard deviation (shaded area) over 5 seeds, evaluated at perturbation radii $\rho = 0.1, 0.2, \ldots, 0.8$. Both methods achieve peak accuracy at $\rho = 0.3$, with 77.05% for SAM-only and 77.49% for the switching strategy.

choice of $\rho$. This is a notable advantage, since the perturbation radius $\rho$ is a difficult hyperparameter to tune in SAM.

The effectiveness of switching can be understood within our theoretical framework. The proof of Theorem 2.1 suggests that hallucinated minimizers tend to arise "near" local maximizers, and hence are more likely to occur in high-loss regions. This further implies that during the early stages of training, when loss values are large, SAM is particularly at risk of entering a basin of attraction of hallucinated minimizers. By first applying gradient descent to escape from high-loss regions before switching to SAM, this risk is mitigated. Consequently, our simple remedy ensures that SAM remains stable even for large perturbation radii.

A related mechanism was studied by Zhou et al. (2025), who applied SAM only in the later stages of training to improve generalization. In contrast, our results show that the switching strategy specifically prevents convergence to hallucinated minimizers, providing a complementary explanation for its effectiveness.

## 5 CONCLUSION

In this work, we identify a previously unrecognized failure mode of SAM: its tendency to converge to hallucinated minimizers. Our theoretical analysis establishes the existence of such minimizers under practical assumptions, and our empirical results validate the theory by demonstrating that SAM can indeed converge to them. To address this, we present a simple switching strategy that effectively avoids hallucinated minimizers.

Although our theoretical and empirical findings are consistent, gaps remain between the theoretical characterizations and broader empirical findings. These gaps open several interesting avenues for follow-up work.

One direction is to extend our theoretical analysis to the setting where SAM employs stochastic gradients rather than full-batch gradients. While our experiments suggest that hallucinated minimizers also arise in the stochastic case, a more rigorous theoretical understanding is desirable. Another direction is to analyze, from a theoretical standpoint, how likely it is for SAM to converge to hallucinated minimizers. Our experiments show that convergence to such minimizers is common, whereas our current theory only guarantees convergence within a local neighborhood of these points. Yet another direction is to extend the analysis to other variants of SAM. Our current results rely on the normalization of the ascent direction, which renders the magnitude of $\nabla f(x)$ irrelevant in constructing hallucinated minimizers. For SAM variants that incorporate gradient magnitude in the ascent step, a modified analysis would be necessary.

## REPRODUCIBILITY STATEMENT

We have taken extensive measures to ensure reproducibility. Complete proofs of all theorems, together with detailed assumptions, are provided in Appendix A and Appendix B. Experimental setups, including datasets and hyperparameters, are described in Appendices C and D. The implementation of our main experiments is provided in the supplementary materials and is also available through an anonymous repository at `https://anonymous.4open.science/r/SAM-can-hallucinate-minimizers-4B82`.

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

# A   OMITTED DETAILS FOR SECTION 2

We begin by showing that the hallucinated minimizers cannot exist when the loss function is convex. For ease of exposition, throughout the appendices we use the following notation: for $x \in \mathbb{R}^d$, let

$$x^+ := x + \rho \frac{\nabla f(x)}{\|\nabla f(x)\|}.$$

**Proposition A.1.** *If the function $f : \mathbb{R}^d \to \mathbb{R}$ is convex, then there is no point $x \in \mathbb{R}^d$ such that* $\nabla f \left( x + \rho \frac{\nabla f(x)}{\|\nabla f(x)\|} \right) = 0.$

*Proof.* Suppose $\nabla f(x) \neq 0$ and $\nabla f(x^+) = 0$. Since $f$ is convex, $x^+$ must be a global minimizer. However, convexity also implies

$$f(x^+) \geq f(x) + \left\langle \nabla f(x), \rho \frac{\nabla f(x)}{\|\nabla f(x)\|} \right\rangle = f(x) + \rho\|\nabla f(x)\| > f(x),$$

which contradicts the optimality of $x^+$.   $\square$

## A.1   FULL PROOF OF THEOREM 2.1

We now provide the full proof of the existence theorem.

**Theorem 2.1.** *Let $f\colon \mathbb{R}^d \to \mathbb{R}$ be continuously differentiable. Assume $f$ has a global minimizer (not necessarily unique) and an isolated local maximizer. Then, a hallucinated minimizer exists for some $\rho > 0$.*

*Proof.* Let $x^\bullet$ be a local maximizer with $f(x^\bullet) = M$, and let $C \subseteq U$ be a compact ball centered at $x^\bullet$ such that $f(x) < f(x^\bullet)$ for all $x \in C$. The strict inequality follows from $x^\bullet$ being an isolated critical point. Define $m := \max_{\partial C} f(x) < M$ and consider the preimage $f^{-1}([M-\varepsilon, M])$ where $0 < \varepsilon < M - m$. Let $C_\varepsilon$ denote the connected component of this preimage containing $x^\bullet \in f^{-1}([M-\varepsilon, M])$.

By Lemma A.2, $C_\varepsilon \subseteq \mathrm{int}\, C$, and hence $C_\varepsilon$ is compact. Moreover, by Lemma A.3, the function value on the boundary satisfies $f(x) = M - \varepsilon$ for all $x \in \partial C_\varepsilon$.

Consider the squared distance function $g(x) = \|x - x_\star\|^2$, and let

$$x_h \in \mathrm{argmax}_{C_\varepsilon} g(x), \quad x_h \neq x_\star, \quad \text{and} \quad \rho := \|x_h - x_\star\|.$$

Then, $x_h$ must be on the boundary of $C_\varepsilon$, and thus $f(x_h) = M - \varepsilon$. Furthermore, since $x^\bullet \neq x_h$ is the only critical point of $f$ in $C_\varepsilon$, the gradient at $x_h$ does not vanish. Consequently, there exists an open neighborhood $V$ of $x_h$ such that $\Sigma := \{x \in V : f(x) = M - \varepsilon\}$ is an embedded $C^1$ hypersurface near $x_h$. By shrinking $V$ if necessary, we may assume $V \cap \partial C_\varepsilon = \Sigma$. Thus, maximizing $g(x)$ over $C_\varepsilon$ is locally equivalent to maximizing $g(x)$ over the hypersurface $\Sigma$. By the method of Lagrange multipliers, we obtain

$$\nabla g(x_h) = \lambda \nabla f(x_h).$$

That is, there exists $\lambda > 0$ such that

$$2(x_\star - x_h) = \lambda \nabla f(x_h)$$

by Lemma A.4. Taking norms of both sides yields $\lambda = \frac{2\rho}{\|\nabla f(x_h)\|}$. Therefore,

$$x_\star = x_h + \rho \frac{\nabla f(x_h)}{\|\nabla f(x_h)\|},$$

which implies that $x_h$ is a hallucinated minimizer.   $\square$

We now prove the three lemmas used in the proof of Theorem 2.1.

**Lemma A.2.** *The set $C_\varepsilon$ from Theorem 2.1 is contained in $\mathrm{int}\, C$. Hence, it is compact.*

*Proof.* Suppose $x \in C_\varepsilon$. If $x \in \partial C$, then $f(x) \geq M - \varepsilon > m$, contradicting the definition of $m$ as the maximum value of $f$ on $\partial C$.

If instead $x \in \text{ext } C = \mathbb{R}^d \setminus C$, then $\text{int } C$ and $\mathbb{R}^d \setminus C$ are two nonempty disjoint open sets that separate $C_\varepsilon$, contradicting the fact that $C_\varepsilon$ is connected. Therefore, $x$ must lie in $\text{int } C$, and $C_\varepsilon \subseteq \text{int } C$. $\qquad\square$

The following lemma shows that every point on $\partial C_\varepsilon$ in Theorem 2.1 lies on the same level set.

**Lemma A.3.** *Any point $x \in \partial C_\varepsilon$ from Theorem 2.1 satisfies $f(x) = M - \varepsilon$.*

*Proof.* Take $x \in \partial C_\varepsilon$. If $f(x) > M - \varepsilon$, continuity of $f$ implies that there exists $r > 0$ such that $f(y) > M - \varepsilon$ for all $y$ in an open ball centered at $x$ with radius $r$. This contradicts the fact that $x$ is a boundary point of $C_\varepsilon$.

If $f(x) < M - \varepsilon$, this directly contradicts the fact that $x \in f^{-1}([M - \varepsilon, M])$. Therefore, $f(x) = M - \varepsilon$. $\qquad\square$

The next lemma establishes that $\lambda > 0$ in Theorem 2.1.

**Lemma A.4.** *In the proof of Theorem 2.1, the vectors $x_\star - x_h$ and $\nabla f(x_h)$ point in the same direction. Equivalently, $\lambda > 0$.*

*Proof.* Let $V$ be an open neighborhood of $x_h$. By possibly shrinking $V$, we may assume the local superlevel set $\{x \in V : f(x) \geq M - \varepsilon\}$ is contained in $C_\varepsilon$. Then, $x_h$ is a maximizer of $g$ over the feasible region $\{x \in V : f(x) \geq M - \varepsilon\}$.

Consider any feasible direction $d \in \mathbb{R}^d$ with $\langle \nabla f(x_h), d \rangle \geq 0$. Let $\gamma \colon (-\varepsilon, \varepsilon) \to \mathbb{R}^d$ be a smooth curve such that $\gamma(0) = x_h$, $\gamma'(0) = d$, and $\gamma(t) \in V$. Then, $f(\gamma(t)) \geq M - \varepsilon$ for sufficiently small $t > 0$. Moreover,

$$\frac{d}{dt} f(\gamma(t)) \bigg|_{t=0} = \langle \nabla f(x_h), d \rangle \geq 0,$$

so $\gamma(t)$ remains in the feasible set for small $t > 0$. Since $x_h$ maximizes $g$, it follows that

$$\lim_{t \downarrow 0} \frac{d}{dt} g(\gamma(t)) = \lim_{t \downarrow 0} \frac{d}{dt} \|\gamma(t) - x_\star\|^2 = \lim_{t \downarrow 0} 2\langle \gamma(t) - x_\star, \gamma'(t) \rangle = 2\langle x_h - x_\star, d \rangle \leq 0.$$

Taking the feasible direction $d = \nabla f(x_h)$ yields

$$2\langle x_h - x_\star, \nabla f(x_h) \rangle = -2\lambda \|\nabla f(x_h)\|^2 \leq 0.$$

Since $\lambda \neq 0$, this inequality implies $\lambda > 0$. $\qquad\square$

### A.2 EXTENDING THEOREM 2.1 TO LOCAL MINIMIZERS

We now extend Theorem 2.1 by relaxing the assumption that $x_\star$ is a global minimizer. In fact, a hallucinated minimizer can still exist when $x_\star$ is only a local minimizer. To show this, we first establish the following lemma.

**Lemma A.5.** *Suppose $\|\nabla f(x) - \nabla f(y)\| \leq L\|x - y\|$ for some $L > 0$, and that $\nabla f(x) \neq 0$ and $\nabla f(y) \neq 0$. Then, we have*

$$\|y + \rho\, u(y) - x - \rho\, u(x)\| \leq \left(1 + \frac{2\rho L}{\|\nabla f(x)\|}\right) \|y - x\|.$$

*Proof.* It follows from the triangle inequality that

$$\|y + \rho\, u(y) - x - \rho\, u(x)\|$$

$$\leq \|y - x\| + \rho\Big\|\frac{\nabla f(y)}{\|\nabla f(y)\|} - \frac{\nabla f(x)}{\|\nabla f(x)\|}\Big\|$$

$$\leq \|y - x\| + \rho\Big\|\frac{\nabla f(y)}{\|\nabla f(y)\|} - \frac{\nabla f(y)}{\|\nabla f(x)\|}\Big\| + \rho\Big\|\frac{\nabla f(y)}{\|\nabla f(x)\|} - \frac{\nabla f(x)}{\|\nabla f(x)\|}\Big\|$$

$$\leq \|y - x\| + \rho\frac{\|\nabla f(x) - \nabla f(y)\|}{\|\nabla f(x)\|} + \rho\frac{\|\nabla f(y) - \nabla f(x)\|}{\|\nabla f(x)\|}$$

$$\leq \|y - x\|\Big(1 + \frac{2\rho L}{\|\nabla f(x)\|}\Big).$$

$\square$

Finally, we obtain the following corollary.

**Corollary A.6.** *Suppose $f$ has a locally Lipschitz gradient. Then, Theorem 2.1 remains valid even when $x_\star$ is a local minimizer of $f$.*

*Proof.* Recall the last equation of the proof of Theorem 2.1:

$$x_\star = x_h + \rho\frac{\nabla f(x_h)}{\|\nabla f(x_h)\|}.$$

Consider an open ball centered at $x_h$ with radius $r$ chosen sufficiently small so that $\nabla f$ does not vanish on the ball. Let $L > 0$ be such that $\|\nabla f(x_h) - \nabla f(y)\| \leq L\|x_h - y\|$ for any $y \in B_r(x_h)$. Since $x_\star$ is a local minimizer, there exists $\delta > 0$ such that

$$f(y) \geq f(x_\star) \quad \forall y \text{ with } \|y - x_\star\| \leq \delta.$$

Now consider an open ball centered at $x_h$ with radius

$$r^\star := \min\Big\{\frac{\delta}{1 + \frac{2\rho L}{\|\nabla f(x_h)\|}}, r\Big\}.$$

Then, for any $y \in B_{r^\star}(x_h)$, we have

$$\|x_\star - y - \rho\, u(y)\| = \|x_h + \rho\, u(x_h) - y - \rho\, u(y)\| \leq \|x_h - y\|\left(1 + \frac{2\rho L}{\|\nabla f(x_h)\|}\right) \leq \delta,$$

where the first inequality follows from Lemma A.5. Hence,

$$f(y + \rho\, u(y)) \geq f(x_\star).$$

This implies $f^{\mathrm{SAM}}(y) \geq f^{\mathrm{SAM}}(x_h)$, so $x_h$ is a local minimizer of $f^{\mathrm{SAM}}$, and therefore a hallucinated minimizer. $\square$

### A.3 MOST MODERN NEURAL NETWORKS ARE REAL-ANALYTIC: FURTHER DISCUSSION

We now formalize the claim that the neural network $\theta \mapsto h_\theta(X)$ is real-analytic in its parameters $\theta$. By standard arguments, the following two lemmas establish that the final loss function

$$f(\theta) = \sum_{i=1}^{N} \ell(h_\theta(X_i), Y_i)$$

is real-analytic under the assumptions on $h_\theta$ stated in Section 2.

**Lemma A.7.** *If $h_\theta \colon \mathbb{R}^d \to \mathbb{R}^h$ is a neural network with real-analytic activation functions, then $h_\theta$ is real-analytic as a function of $\theta$.*

*Proof.* Fix input data $x$ and let $z$ denote the hidden states, which depend on $x$ and $\theta$. Then, $z$, hence the entire network $h_\theta$, is a finite composition of the following real-analytic mappings:

- (Layer normalization) $z \mapsto \gamma \odot \frac{z - \mu_L(z)}{\sqrt{\sigma_L(z) + \varepsilon}} + \beta$, where $\mu_L(z)$ and $\sigma_L(z)$ are the per-sample mean and variance;

- (Batch normalization) $z \mapsto \gamma \odot \frac{z - \mu_B(z)}{\sqrt{\sigma_B(z) + \varepsilon}} + \beta$, where $\mu_B(z)$ and $\sigma_B(z)$ are the per-channel mean and variance;

- (Activation) $z \mapsto \sigma(z)$, where $\sigma$ is a real-analytic activation function;

- (Softmax) $z \mapsto \mu(z)$, where $\mu$ is the softmax function;

- (Average pooling) $z \mapsto Az$, where $A$ is a linear averaging operator;

- (Residual connection) $z \mapsto z + F_\theta(z)$, where $F$ is real-analytic in $\theta$;

- (Convolution layer) $(z, W_i, b_i) \mapsto W_i * z + b_i$, where $*$ denotes the discrete convolution operator;

- (Dropout) $z \mapsto m \odot z$, where $m$ is a masking operator;

- (Linear layer) $(z, A_i, b_i) \mapsto A_i z + b_i$; and

- (Attention layer) $(z, W_Q, W_K, W_V) \mapsto \mu\left(\frac{QK^\top}{\sqrt{d_{\mathrm{attn}}}}\right)V$, where $Q = zW_Q$, $K = zW_K$, $V = zW_V$, and $d_{\mathrm{attn}}$ is the size of the attention matrix $Q, K$.

Since the composition of real-analytic functions is real-analytic, it follows that $h_\theta$ is real-analytic. $\square$

Finally, the next lemma establishes that $f$ is real-analytic. This follows directly from the fact that the composition of real-analytic functions is real-analytic; hence the proof is omitted.

**Lemma A.8.** *Let $h_\theta$ be a (finite) neural network constructed with linear layers, attention, convolution, layer normalization, and real-analytic activation functions (all commonly used activation functions except ReLU are real-analytic). Then, for a real-analytic loss function $\ell$,*

$$f(\theta) := \frac{1}{N} \sum_{i=1}^{N} \ell(h_\theta(X_i), Y_i)$$

*is real-analytic as a function of $\theta$.*

## A.4 FULL PROOF OF THEOREM 2.2

The following lemma shows that, under the real-analyticity assumption, critical points cannot accumulate around the local maximizer set $X$. The argument relies on the Łojasiewicz inequality.

**Lemma A.9.** *Suppose $f \colon \mathbb{R}^d \to \mathbb{R}$ is real-analytic and $X$ is a bounded local maximizer set of $f$ with some $\delta > 0$. Then, there exists $\varepsilon > 0$ with the following property: if $x$ is a critical point in the $\delta$-neighborhood of $X$ with $f(x) \geq f(X) - \varepsilon$, then $x \in X$.*

*Proof.* Define the closed $\delta$-neighborhood of $X$ by $\mathcal{N}_\delta(X) := \{y : d(y, X) \leq \delta\}$. Since $X$ is a bounded connected set, $\mathcal{N}_\delta(X)$ is compact and connected. Let $S$ denote the set of critical points in $\mathcal{N}_\delta(X)$ that are not in $X$:

$$S = \{s \in \mathcal{N}_\delta(X) : \nabla f(s) = 0\} \setminus X.$$

If $S = \emptyset$, then the theorem holds for any $\varepsilon > 0$, and we are done. Assume instead that $S \neq \emptyset$. For each point $x \in X$, Lemma 2.3 guarantees the existence of an open neighborhood $U_x$, a constant $C_x > 0$, and an exponent $q_x \in (0, 1)$ such that

$$|f(x) - f(y)|^{q_x} = |f(X) - f(y)|^{q_x} \leq C_x \|\nabla f(y)\|, \quad y \in U_x.$$

If $y \in S$, then

$$|f(X) - f(y)|^{q_x} \leq C_x \|\nabla f(y)\| = 0.$$

This implies $f(y) = f(X)$, which is a contradiction. Hence, $S \subseteq C := \mathcal{N}_\delta(X) \setminus \bigcup_{x \in X} U_x$. (Note that $C \neq \emptyset$ since $S \neq \emptyset$). Since $C$ is compact, define

$$0 < \varepsilon^\star := f(X) - \max_{y \in C} f(y).$$

Then, any $\varepsilon \in (0, \varepsilon^\star)$ satisfies the theorem. $\qquad\square$

Finally, we use Lemma A.9 to complete the proof of Theorem 2.2.

**Theorem 2.2.** *Let $f \colon \mathbb{R}^d \to \mathbb{R}$ be real-analytic. Assume $f$ has a global minimizer (not necessarily unique) and a bounded local maximizer set. Then, a hallucinated minimizer exists for some $\rho > 0$.*

*Proof.* Define the closed $\delta$-neighborhood of a local maximizer set $X$ as $\mathcal{N}_\delta(X) := \{y : d(y, X) \leq \delta\}$. Since $X$ is a bounded connected set, $\mathcal{N}_\delta(X)$ is compact and connected. By Lemma A.9, there exists $\varepsilon_1 > 0$ such that any critical point in $\mathcal{N}_\delta(X)$ with function value at least $f(X) - \varepsilon_1$ must lie in $X$. Next, choose $\varepsilon_2 > 0$ such that $0 < \varepsilon_2 < f(X) - \max_{\partial \mathcal{N}_\delta(X)} f(x)$.

Let $\varepsilon := \min\{\varepsilon_1, \varepsilon_2\}$, and consider the preimage $f^{-1}([f(X) - \varepsilon, f(X)])$. Define $C_\varepsilon$ as the connected component of this preimage that contains $X$.

By the same reasoning as in Lemma A.2, $C_\varepsilon \subseteq \operatorname{int} \mathcal{N}_\delta(X)$, and hence $C_\varepsilon$ is compact. Moreover, by Lemma A.3, every point $x \in \partial C_\varepsilon$ satisfies $f(x) = f(X) - \varepsilon$.

Now define $g(x) = \|x - x_\star\|^2$ and let

$$x_h \in \operatorname{argmax}_{C_\varepsilon} g(x), \quad x_h \neq x_\star, \quad \rho := \|x_h - x_\star\|.$$

Since the only critical points in $C_\varepsilon$ are those in $X$, we have $\nabla f(x_h) \neq 0$. Thus, there exists an open neighborhood $V$ of $x_h$ such that $\Sigma := \{x \in V : f(x) = f(X) - \varepsilon\}$ is an embedded smooth hypersurface near $x_h$. By possibly shrinking $V$, we may assume $V \cap \partial C_\varepsilon = \Sigma$. Therefore, maximizing $g(x)$ over $C_\varepsilon$ is locally equivalent to maximizing $g(x)$ over the hypersurface $\Sigma$.

Then, by the method of Lagrange multipliers, we obtain

$$\nabla g(x_h) = \lambda \nabla f(x_h),$$

which implies that there exists $\lambda > 0$ such that

$$2(x_\star - x_h) = \lambda \nabla f(x_h).$$

The positivity of $\lambda$ follows from the same reasoning as in Lemma A.4. Taking norms of both sides yields $\lambda = \frac{2\rho}{\|\nabla f(x_h)\|}$. Therefore,

$$x_\star = x_h + \rho \frac{\nabla f(x_h)}{\|\nabla f(x_h)\|},$$

which shows that $x_h$ is a hallucinated minimizer. $\qquad\square$

# B    OMITTED DETAILS FOR SECTION 3

## B.1    PROOF OF THEOREM 3.1

In this subsection, we prove Theorem 3.1. The argument relies on the implicit function theorem, which we state below.

**Theorem B.1** (Implicit function theorem, Lee (2013)). *Let $U \subseteq \mathbb{R}^d \times \mathbb{R}^d$ be an open set, and let $(x, y)$ denote the coordinates on $U$. Suppose $\Phi : U \to \mathbb{R}^d$ is a smooth function, $(a, b) \in U$, and $c = \Phi(a, b)$. If the $d \times d$ matrix*

$$\left( \frac{\partial \Phi^i}{\partial y^j}(a, b) \right)$$

*is invertible, then there exist neighborhoods $V_0 \subseteq \mathbb{R}^d$ of $a$ and $W_0 \subseteq \mathbb{R}^d$ of $b$, together with a smooth function $F : V_0 \to W_0$, such that $\Phi^{-1}(c) \cap (V_0 \times W_0)$ is the graph of $F$. In other words, $\Phi(x, y) = c$ for $(x, y) \in V_0 \times W_0$ if and only if $y = F(x)$.*

Now we prove Theorem 3.1.

**Theorem 3.1.** *Suppose* $f : \mathbb{R}^d \to \mathbb{R}$ *satisfies the assumptions of Theorem 2.2. Assume* $\mathcal{M} \subseteq$ $\operatorname{argmin} f$, *where* $\mathcal{M} \subseteq \mathbb{R}^d$ *is a nonempty smooth* $m$-dimensional manifold. Let $x_h$ *be a hallucinated minimizer with a corresponding* $\rho > 0$ *as constructed in the proof of Theorem 2.2. If* $I + \rho \nabla u(x_h) \in \mathbb{R}^{d \times d}$ *is nonsingular, then the set of hallucinated minimizers contains a smooth manifold of dimension* $m$.

*Proof.* Let $x_h$ be a hallucinated minimizer of $f$ constructed in the proof of Theorem 2.2. Then, it satisfies

$$x_\star = x_h + \rho \frac{\nabla f(x_h)}{\|\nabla f(x_h)\|} = x_h + \rho u(x_h)$$

for some perturbation radius $\rho > 0$. Let $V$ be an open neighborhood of $x_h$ on which the gradient never vanishes. Such a neighborhood exists because $\|\nabla f(x_h)\| > 0$ and $\nabla f$ is continuous. Define $F : \mathbb{R}^d \times V \to \mathbb{R}^d$ by

$$F(x, y) = y + \rho\, u(y) - x.$$

Clearly, $F(x_\star, x_h) = 0$, and by assumption, $\frac{dF}{dy}(x_\star, x_h) = I + \rho \nabla u(x_h)$ is invertible. By the implicit function theorem, there exist open neighborhoods $U_0 \subseteq \mathbb{R}^d$ of $x_\star$ and $V_0 \subseteq V$ of $x_h$, together with a smooth map $G : U_0 \to V_0$, such that $G(x_\star) = x_h$ and

$$F(x, G(x)) = 0 \quad \forall x \in U_0.$$

This also implies that $G$ is also a local diffeomorphism at $x_\star$, since the differential is invertible at $x_\star$:

$$-I + \frac{dF}{dy}\bigg|_{y=x_h} \frac{dG}{dx}\bigg|_{x=x_\star} = 0 \iff \frac{dG}{dx}\bigg|_{x=x_\star} = \left(\frac{dF}{dy}\right)^{-1}\bigg|_{y=x_h}.$$

In particular, there exists an open $U \subseteq U_0$ around $x_\star$ such that $G|_U : U \to G(U)$ is a diffeomorphism. Since $U \cap \mathcal{M}$ is an $m$-dimensional manifold (without boundary), the image $G(U \cap \mathcal{M})$ under the diffeomorphism is also an $m$-dimensional manifold.

Finally, note that any $y \in G(U \cap \mathcal{M})$ satisfies

$$x = y + \rho \frac{\nabla f(y)}{\|\nabla f(y)\|},$$

where $x = G^{-1}(y) \in \mathcal{M} \cap U$ and hence $x \in \arg\min f$. Thus, $G(U \cap \mathcal{M})$ forms an $m$-dimensional manifold of hallucinated minimizers. $\qquad\square$

### B.2 Proof of Theorem 3.2 and further discussion

In this subsection, we prove Theorem 3.2 and then discuss the special case of isolated hallucinated minimizers.

**Theorem 3.2.** *Suppose* $f : \mathbb{R}^d \to \mathbb{R}$ *is real-analytic, and let* $H \subset \mathbb{R}^d$ *be a bounded, connected set of hallucinated minimizers of* $f$ *for a fixed perturbation radius* $\rho > 0$. *Assume there exists* $\delta > 0$ *such that the* $\delta$-neighborhood of $H$ contains no minimizers of $f^{\mathrm{SAM}}$ other than those already in $H$. *Assume further that every* $x_h \in H$ *satisfies*

$$1 + \rho \lambda_{\min}(\mathrm{Sym}(\nabla u(x_h))) > 0, \quad \text{where } \mathrm{Sym}(\nabla u(x_h)) = \frac{1}{2}(\nabla u(x_h) + \nabla u(x_h)^\top).$$

*If the initialization* $x_0$ *is chosen sufficiently close to* $H$, *then there exists a sufficiently small fixed step size* $\eta_k = \eta > 0$ *such that the SAM iterates converge to* $H$, *in the sense that* $d(x_k, H) \to 0$.

*Proof.* First, we claim that the set $H$ is closed (hence compact) by construction. Indeed, if $x_h \in \bar{H}$, then the corresponding function value must equal $f^{\mathrm{SAM}}(H) = \min f^{\mathrm{SAM}}$. By our assumption on $H$, this implies $x_h \in H$. Hence $H = \bar{H}$.

Let $\mathcal{N}_\delta(H)$ denote the closed $\delta$-neighborhood of $H$ from the theorem assumption. Since $1 + \rho \lambda_{\min}(\mathrm{Sym}(\nabla u(x_h))) > 0$ and $\|\nabla f(x_h)\| > 0$ for all $x_h \in H$, there exists an open neighborhood $W$ of $H$ such that $1 + \rho \lambda_{\min}(\mathrm{Sym}(\nabla u(x))) > 0$ and $\|\nabla f(x)\| > 0$ for any $x \in W$. By shrinking $\mathcal{N}_\delta(H)$ if necessary, we may assume $\mathcal{N}_\delta(H) \subseteq W$ and $f^{\mathrm{SAM}}$ is real-analytic on $\mathcal{N}_\delta(H)$.

Applying an argument analogous to Lemma A.9, with local maximizers replaced by minimizers, we obtain $\varepsilon^\star > 0$ such that if $x$ is a critical point in $\mathcal{N}_\delta(H)$ with $f^{\mathrm{SAM}}(x) \leq f^{\mathrm{SAM}}(H) + \varepsilon^\star$, then $x \in H$.

Now consider the closed neighborhood $\mathcal{N}_{\delta/2}(H)$, and set

$$m := \min_{x \in \partial \mathcal{N}_{\delta/2}(H)} f^{\mathrm{SAM}}(x) > f_\star,$$

where $f_\star = f^{\mathrm{SAM}}(H)$. The strict inequality follows from the construction of $\mathcal{N}_\delta(H)$. Choose $\varepsilon > 0$ such that

$$0 < \varepsilon < m - f^{\mathrm{SAM}}(H) = m - f_\star \quad \text{and} \quad 0 < \varepsilon < \varepsilon^\star.$$

Let $C_\varepsilon$ be the connected component of the sublevel set

$$(f^{\mathrm{SAM}})^{-1}((-\infty, f_\star + \varepsilon]) = (f^{\mathrm{SAM}})^{-1}([f_\star, f_\star + \varepsilon])$$

that contains $H$. Then, $C_\varepsilon$ is compact and contains no other critical points of $f^{\mathrm{SAM}}$ besides those in $H$. The proof of $C_\varepsilon$ being bounded (hence compact) by $\mathcal{N}_{\delta/2}(H)$ is analogous to Lemma A.2.

Define

$$C_\rho := \{x : d(x, C_\varepsilon) \leq \rho\}.$$

Then, $C_\rho$ is also compact. Set

$$M := \max_{x \in C_\rho} \|\nabla f(x)\| > 0, \qquad L := \max_{x \in \mathcal{N}_\delta(H)} \|\nabla^2 f(x)\| > 0,$$

and

$$\gamma := \min_{x \in \mathcal{N}_\delta(H)} (1 + \rho \lambda_{\min}(\mathrm{Sym}(\nabla u(x)))) > 0.$$

Consider the SAM update with fixed $\rho > 0$ and constant step size $\eta_k = \eta$ chosen such that

$$0 < \eta < \min\left\{\frac{\delta}{2M}, \frac{2\gamma}{L}\right\},$$

and initialization at $x_0 \in C_\varepsilon$. We show by induction that the SAM iterates $\{x_k\}$ remain in $C_\varepsilon$. The base case $x_0 \in C_\varepsilon$ is true by assumption. Suppose $x_k \in C_\varepsilon$. Then, by definition of the SAM update, $x_k^+ \in C_\rho$ and $x_{k+1} = x_k - \eta \nabla f(x_k^+)$. We claim $x_{k+1} \in \mathcal{N}_\delta(H)$, since

$$\begin{aligned}
d(x_{k+1}, H) &= \inf_{x_h \in H} \|x_{k+1} - x_h\| \\
&\leq \inf_{x_h \in H} \|x_k - x_h\| + \|x_{k+1} - x_k\| \\
&\leq \frac{\delta}{2} + \eta \|\nabla f(x_k^+)\| \\
&\leq \frac{\delta}{2} + \frac{\delta}{2M} \cdot M \\
&\leq \delta.
\end{aligned}$$

By $L$–smoothness of $f^{\mathrm{SAM}}$ on $\mathcal{N}_\delta(H)$,

$$\begin{aligned}
f^{\mathrm{SAM}}(x_{k+1}) &\leq f^{\mathrm{SAM}}(x_k) + \langle \nabla f^{\mathrm{SAM}}(x_k),\, x_{k+1} - x_k \rangle + \frac{L}{2} \|x_{k+1} - x_k\|^2 \\
&= f^{\mathrm{SAM}}(x_k) - \eta \langle \nabla f^{\mathrm{SAM}}(x_k), \nabla f(x_k^+) \rangle + \frac{L\eta^2}{2} \|\nabla f(x_k^+)\|^2.
\end{aligned}$$

Since

$$\nabla f^{\mathrm{SAM}}(x) = (I + \rho \nabla u(x)) \nabla f(x^+),$$

we obtain

$$f^{\text{SAM}}(x_{k+1}) \leq f^{\text{SAM}}(x_k) - \eta \langle \nabla f^{\text{SAM}}(x_k), \nabla f(x_k^+) \rangle + \frac{L\eta^2}{2} \|\nabla f(x_k^+)\|^2$$

$$= f^{\text{SAM}}(x_k) - \eta\rho\langle \nabla u(x_k)\nabla f(x_k^+), \nabla f(x_k^+) \rangle - \eta\|\nabla f(x_k^+)\|^2 + \frac{L\eta^2}{2}\|\nabla f(x_k^+)\|^2$$

$$\leq f^{\text{SAM}}(x_k) - \eta\rho\lambda_{\min}(\text{Sym}(\nabla u(x_k)))\|\nabla f(x_k^+)\|^2 - \eta\|\nabla f(x_k^+)\|^2 + \frac{L\eta^2}{2}\|\nabla f(x_k^+)\|^2$$

$$\leq f^{\text{SAM}}(x_k) - \eta\left(1 + \rho\lambda_{\min}(\text{Sym}(\nabla u(x_k))) - \frac{L\eta}{2}\right)\|\nabla f(x_k^+)\|^2$$

$$\leq f^{\text{SAM}}(x_k) - \eta\left(\gamma - \frac{L\eta}{2}\right)\|\nabla f(x_k^+)\|^2$$

$$< f^{\text{SAM}}(x_k),$$

where the last inequality follows from $0 < \eta < \frac{2\gamma}{L}$. Moreover, since the descent property $f^{\text{SAM}}(x_{k+1}) < f^{\text{SAM}}(x_k)$ holds when $\eta$ is replace by $\eta t$ for $t \in [0, 1]$, the line segment from $x_k$ to $x_{k+1}$ lies within the sublevel set $(f^{\text{SAM}})^{-1}([f_\star, f_\star + \varepsilon])$. Thus, because $x_k \in C_\varepsilon$, we conclude $x_{k+1} \in C_\varepsilon$ by the connectedness of $C_\varepsilon$. This completes the induction.

Finally, since $f^{SAM}(x_k)$ is decreasing and bounded below, we have $\eta\left(\gamma - \frac{L\eta}{2}\right)\|\nabla f(x_k^+)\| \to 0$. Hence, $\nabla f(x_k^+) \to 0$. If $x_\infty$ is a limit point of $\{x_k\}_{k=0,1,\ldots}$, then $\nabla f(x_\infty^+) = 0$. By construction of $C_\varepsilon$, this implies $x_\infty \in H$. Therefore, $d(x_k, H) \to 0$. $\qquad\square$

To discuss point convergence to isolated hallucinated minimizers, we now turn to the case where the manifold $\mathcal{M}$ in Theorem 3.1 reduces to a single isolated point, i.e., 0-dimensional. In this case, we can show that the corresponding hallucinated minimizer is also isolated. This can be viewed as the special case where $H$ in Theorem 3.2 is a singleton.

**Lemma B.2.** *Let $x_\star$ be a minimizer of a $C^1$ function $f : \mathbb{R}^d \to \mathbb{R}$ satisfying the assumptions of Theorem 2.1. Suppose $x_\star$ is an isolated minimizer; that is, there exists an open neighborhood of $x_\star$ in which it is the unique critical point and the unique minimizer. Let $x_h$ be a hallucinated minimizer of $f$ constructed in the proof of Theorem 2.1 with $\rho > 0$. If $I + \rho\nabla u(x_h)$ is invertible, then there exists an open neighborhood $W$ of $x_h$ such that*

- *$\nabla f$ never vanishes on $W$, and thus $f^{\text{SAM}}$ is well-defined on $W$;*

- *no point other than $x_h$ satisfies $\nabla f(x^+) = 0$ in $W$; and*

- *$x_h$ is the unique hallucinated minimizer in $W$.*

Such a point $x_h$ is called an *isolated hallucinated minimizer*.

*Proof.* Since $x_h$ is a hallucinated minimizer of $f$ constructed in the proof of Theorem 2.1, it satisfies

$$x_\star = x_h + \rho\frac{\nabla f(x_h)}{\|\nabla f(x_h)\|} = x_h + \rho u(x_h)$$

for some perturbation radius $\rho > 0$. Let $U_0$ and $V_0$ be open neighborhoods of $x_\star$ and $x_h$, respectively, as constructed in the proof of Theorem 3.1. Also, let $U_1$ be an open neighborhood of $x_\star$ that contains no other minimizers, by assumption. Define $W := G(U_0 \cap U_1)$, where $G$ is the $C^1$ mapping constructed in the proof of Theorem 3.1. We claim that $W$ is the desired open neighborhood of $x_h$.

First, $W$ is open since $G$ is a local diffeomorphism at $x_\star$. Moreover, $f^{\text{SAM}}$ is well-defined on $W$ by the construction of $V_0$. Suppose $y \in W$ satisfies $\nabla f(y^+) = 0$. Then, since

$$x = y + \rho\frac{\nabla f(y)}{\|\nabla f(y)\|}$$

for the unique $x = G^{-1}(y) \in U_1$, it follows that $\nabla f(y^+) = \nabla f(x) = 0$, contradicting the fact that $x$ is an isolated minimizer. Similarly, if $y$ is a hallucinated minimizer, then the uniqueness of $x = G^{-1}(y)$ implies $x = x_\star$, and hence $y = x_h$. This proves the claim. $\qquad\square$

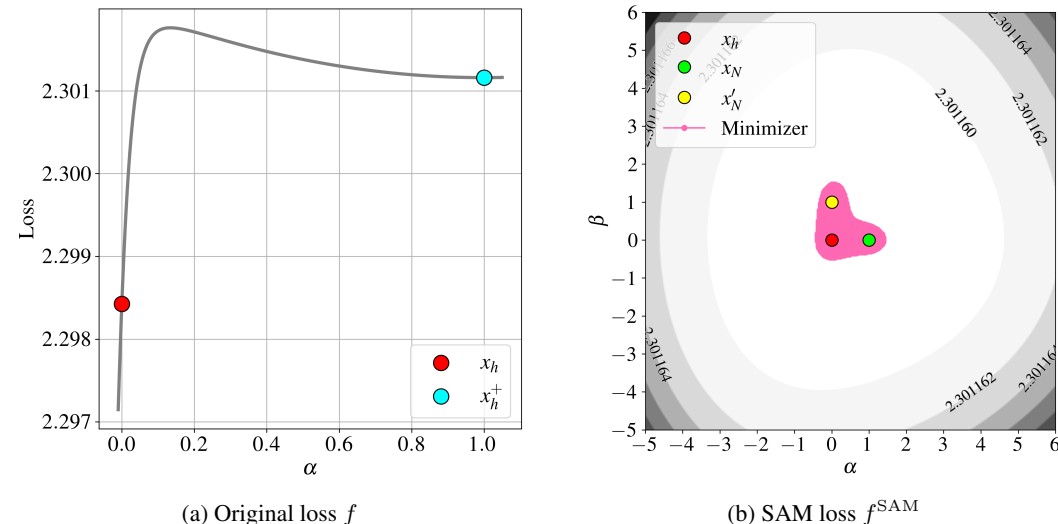

(a) Original loss $f$  (b) SAM loss $f^{\text{SAM}}$

Figure 7: Visualizations of the hallucinated minimizer $x_h$: (a) original loss $f$ along the line between $x_h$ and $x_h^+$; (b) SAM loss $f^{\text{SAM}}$ over the plane defined by $x_h$, $x_N$, and $x_N'$.

Then, as a corollary of Theorem 3.2 and the previous lemma, we obtain the following point convergence result.

**Corollary B.3.** *Assume $f$ satisfies the assumptions in Theorem 2.1 and, in addition, $f \in C^2$. Let $x_h$ be an isolated hallucinated minimizer of $f$ for a fixed perturbation radius $\rho > 0$, constructed in the proof of Theorem 2.1. Suppose*

$$1 + \rho\lambda_{\min}(\text{Sym}(\nabla u(x_h))) > 0.$$

*Then, the SAM iterates, when initialized sufficiently close to $x_h$, converge to $x_h$ for sufficiently small fixed step size $\eta_k = \eta$.*

The real-analytic property of $f$ is used to apply the Łojasiewicz inequality in order to construct a neighborhood where critical points do not accumulate. However, since Lemma B.2 guarantees the existence of isolated hallucinated minimizers without this assumption, the real-analytic condition is not required here. Hence, the $C^2$ assumption on $f$ is sufficient to ensure the existence of constants $L$ and $\gamma$ as in the proof of Theorem 3.2.

## C  EXPERIMENTAL DETAILS FOR SAM IN DEEP LEARNING

### C.1  SAM WITH FULL-BATCH GRADIENTS

In Section 4.1, we train a neural network using SAM with full-batch gradients. Specifically, the model is a two-layer network with 128 hidden units and `Tanh` activations, trained on the MNIST dataset (LeCun et al., 1998). The classification task uses cross-entropy loss. Training is implemented in PyTorch (Paszke et al., 2019) with a learning rate of 0.01, momentum 0.9, and no weight decay. We run 20 million updates with perturbation radius $\rho = 1.8$ to obtain the convergence point $x_h$, whose loss landscape is shown in Figure 4. In this subsection, we provide additional visualizations to further examine the local properties of $x_h$.

Figure 7(a) presents a one-dimensional view along the line connecting $x_h$ and $x_h^+ = x_h + \rho\frac{\nabla f(x_h)}{\|\nabla f(x_h)\|}$, parameterized as $x(\alpha) = (1 - \alpha)x_h + \alpha x_h^+$. The plot shows that $x_h$ is not a minimizer of the original objective and that the surrounding loss landscape differs substantially from that around $x_h^+$. This demonstrates that the phenomenon of SAM converging to a hallucinated minimizer is fundamentally distinct from the case in which a saddle point becomes an attractor, which requires the surrounding quadratic structure to hold (Compagnoni et al., 2023).

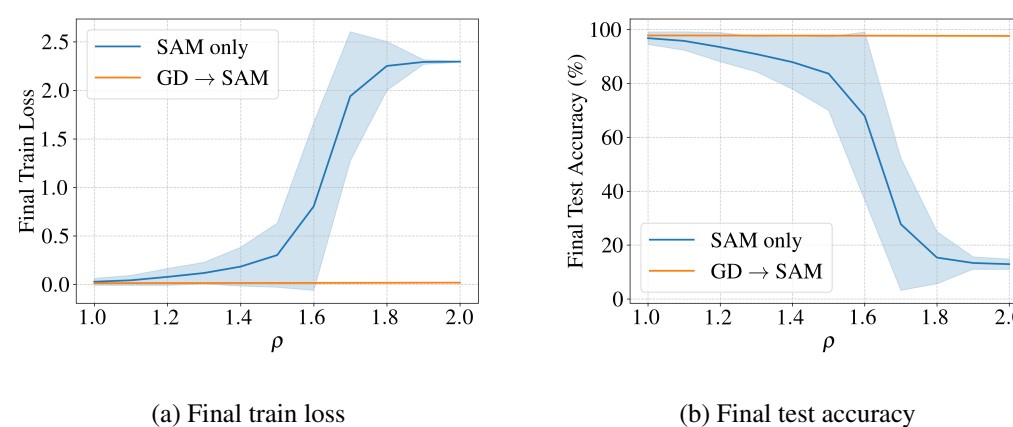

(a) Final train loss                    (b) Final test accuracy

Figure 8: Comparison of SAM-only and the switching strategy across different perturbation radii $\rho$. Results are obtained using SAM with full-batch gradients over 80 seeds. Bold lines indicate the mean, and shaded areas represent the standard deviation.

Figure 7(b) extends Figure 4(c) with an additional visualization of the SAM loss. We initialize $x_0$ by adding a small random perturbation of magnitude 0.1 to $x_h$, and then perform $N = 1000$ SAM steps, yielding the same $x_N$ reported in Figure 4(c). Applying an independent perturbation followed by the same procedure gives $x'_N$. We then consider the plane spanned by $x_h$, $x_N$, and $x'_N$, parameterized as $x(\alpha, \beta) = x_h + \alpha u + \beta v$ with $u = x_N - x_h$ and $v$ chosen orthogonal to $u$. On this plane, the visualization shows that the hallucinated minimizers are not confined to a one-dimensional curve but instead extend into a two-dimensional surface-like structure.

In the experiments reported in Figure 5, we investigate SAM with full-batch gradients by varying both the perturbation radius and the random seeds. Under the same experimental setting, Figure 8 shows the final training loss and test accuracy at the last step for perturbation radii $\rho = 1.0, 1.1, \ldots, 2.0$, evaluated across 80 seeds. The results demonstrate that the performance of SAM is highly sensitive to the perturbation radius, whereas the switching strategy maintains stable performance even for larger values of $\rho$.

### C.2 SAM WITH STOCHASTIC GRADIENTS

We examine whether the phenomena observed with full-batch SAM also arise in the stochastic setting, as shown in Figure 6. ResNet-18 is trained on CIFAR-100 with standard data augmentations, including random cropping with padding, horizontal flipping, and Cutout (DeVries & Taylor, 2017). The mini-batch size is 64, the learning rate 0.01, momentum 0.9, and weight decay $10^{-4}$. Training proceeds for 200 epochs with cosine-annealed learning rates (Loshchilov & Hutter, 2016), following the practical implementation of FSAM (Li et al., 2024). The switching strategy applies plain stochastic gradient descent for the first 10% of epochs before switching to stochastic SAM.

Under the same setting, Figure 9 reports the CIFAR-100 results, comparing SAM-only with the switching strategy. Experiments are conducted for perturbation radii $\rho = 0.1, 0.4, 0.7, 1.0$ across 16 random seeds. Each curve shows training loss and test accuracy over epochs, with bold lines denoting the mean across seeds and shaded regions indicating the standard deviation. The results show that, as in the full-batch case, SAM performance degrades with larger perturbation radii, whereas the switching strategy remains stable and robust across all settings.

### D TWO-DIMENSIONAL SYNTHETIC FUNCTION FOR VISUALIZATION

To visualize how the SAM perturbation radius $\rho$ affects the objective, we introduce the following two-dimensional synthetic function (originally illustrated in Figure 1):

$$f(x,y) = 0.8 \exp\left(-\frac{x^2 + y^2}{(2.5)^2}\right) \cdot W_X(x) - \exp\left(-(x + 1.55\cos(y/1.5))^2\right) \cdot W_Y(y) + 1,$$

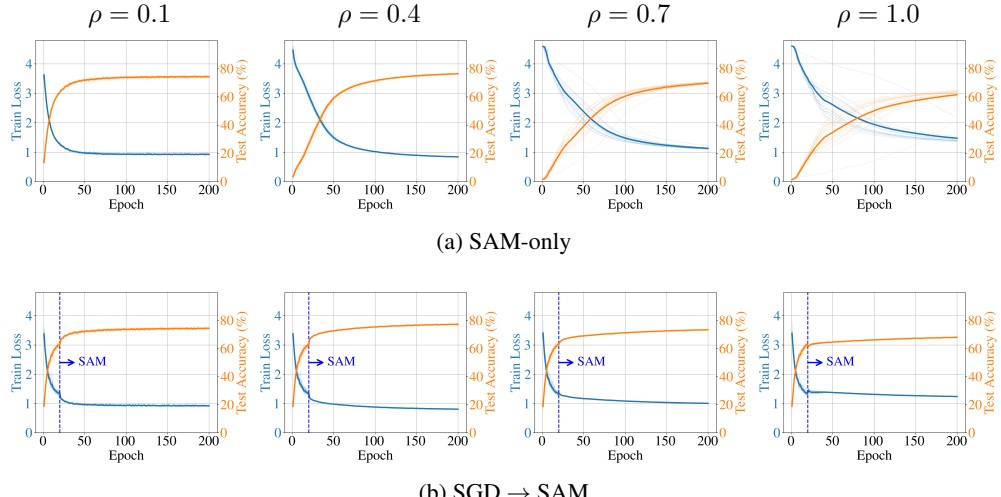

Figure 9: Training loss and test accuracy curves for SAM with stochastic gradients on CIFAR-100 using ResNet-18. The top row shows SAM-only training, while the bottom row applies a switching strategy that runs stochastic gradient descent for the first 10% of epochs before switching to SAM. Columns correspond to perturbation radii $\rho \in \{0.1, 0.4, 0.7, 1.0\}$. Final average test accuracies are 74.29%, 76.36%, 69.64%, and 61.58% for SAM-only, compared to 74.48%, 77.22%, 73.35%, and 67.93% for the switching strategy.

where

$$W_X(x) = \begin{cases} 0, & x \leq -1, \\ 0.5\left(1 - \cos(\pi(x+1))\right), & -1 < x < 0, \\ 1, & x \geq 0, \end{cases}$$

$$W_Y(y) = \begin{cases} 1, & |y| \leq 0.6, \\ 0.5\left(1 + \cos\left(\pi \cdot \frac{|y|-0.6}{5.0}\right)\right), & 0.6 < |y| < 5.6, \\ 0, & |y| \geq 5.6. \end{cases}$$

The function $f(x, y)$ is continuously differentiable, since all its components are smoothly joined. It is designed so that its minimizer set forms a curve. In fact, the global minimizer set of $f$ is exactly

$$\left\{(x, y) \in \mathbb{R}^2 \;\middle|\; x = -1.55\cos\left(\tfrac{y}{1.5}\right), \; |y| \leq 0.6\right\}.$$

Figure 1 shows the original function $f$, the SAM objective $f^{\mathrm{SAM}}$, and its gradient $\nabla f(x + \rho \frac{\nabla f(x)}{\|\nabla f(x)\|})$ at perturbation radius $\rho = 2.8$. To further examine the effect of the perturbation radius, Figures 10 and 11 illustrate how the SAM objective $f^{\mathrm{SAM}}$ and its gradient $\nabla f(x + \rho \frac{\nabla f(x)}{\|\nabla f(x)\|})$ evolve as $\rho$ varies over the range $0, 0.5, \ldots, 3.5$. In this setting, the SAM minimizers are defined as the regions where $f^{\mathrm{SAM}}$ attains its minimum values; in practice, they appear either as isolated points or as continuous curve-like structures.

An analysis of the SAM minimizers as a function of the perturbation radius $\rho$ reveals two distinct regimes. For small $\rho$, the minimizers approach the critical points of $f$. Although $f^{\mathrm{SAM}}$ is not defined at a critical point, higher-resolution numerical experiments show convergence arbitrarily close to such points. At $\rho = 0$, $f^{\mathrm{SAM}}$ coincides with $f$, and the minimizers exactly match those of $f$. For $\rho = 0.5$, the minimizers remain on the original minimizer set, whereas for $\rho = 1.0$ and $\rho = 1.5$, they shift toward the maximizers of $f$. The corresponding gradient fields indicate that these critical points act as attractors of the SAM dynamics, consistent with the theoretical analysis of Compagnoni et al. (2023).

In contrast, for larger perturbation radii, the SAM minimizers form a curve on the right-hand side, starting near the maximizer and drifting outward as $\rho$ increases. This behavior is consistent with the proof of Theorem 2.1, which shows that hallucinated minimizers emerge for a sufficiently large

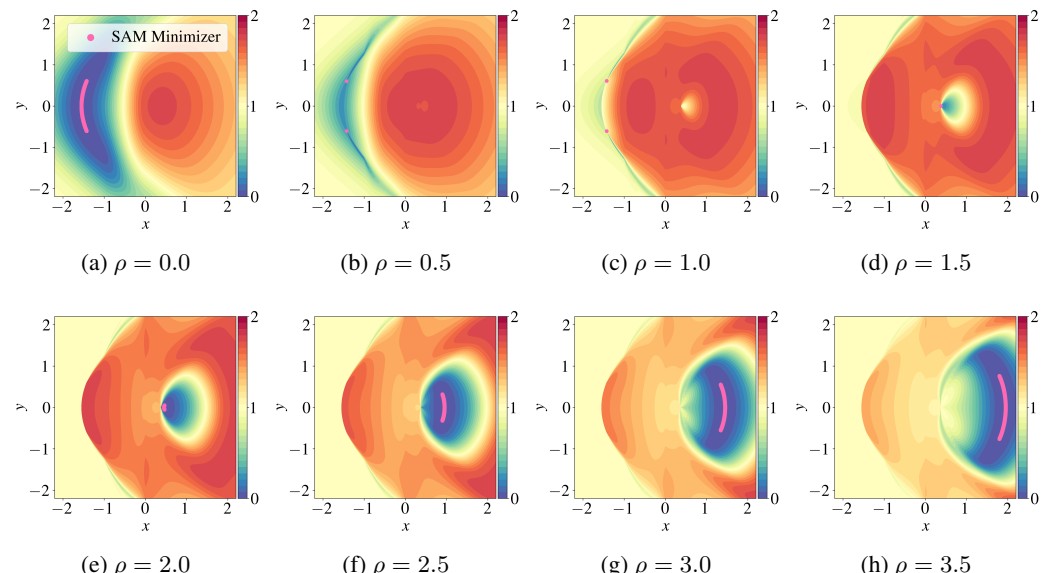

Figure 10: SAM objective $f^{\mathrm{SAM}}(x)$ under different perturbation radii $\rho$. The corresponding SAM minimizers are shown in pink.

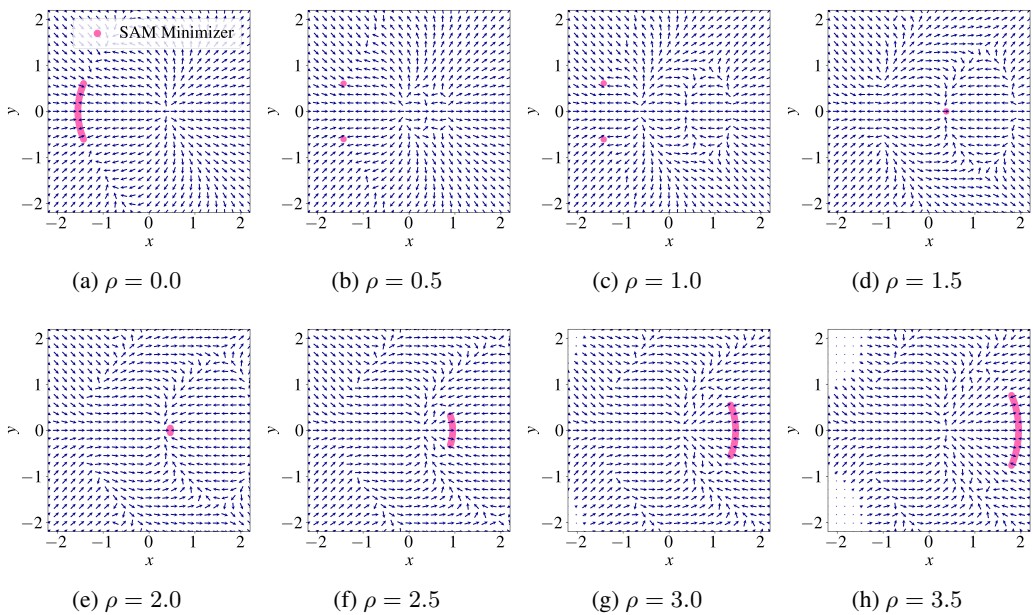

Figure 11: SAM gradient field $\nabla f(x + \rho \frac{\nabla f(x)}{\|\nabla f(x)\|})$ for different perturbation radii $\rho$. The corresponding SAM minimizers are shown in pink.

perturbation radius $\rho$. Theorem 3.1 further establishes that these minimizers preserve the dimensionality of the original minimizer manifold. Meanwhile, the SAM gradient field shows that these minimizers act as attractors within their neighborhood, and the conditions of Theorem 3.2 are indeed satisfied. Taken together, these observations show that our theory offers a full theoretical explanation of the empirical phenomenon of hallucinated minimizers in this example.

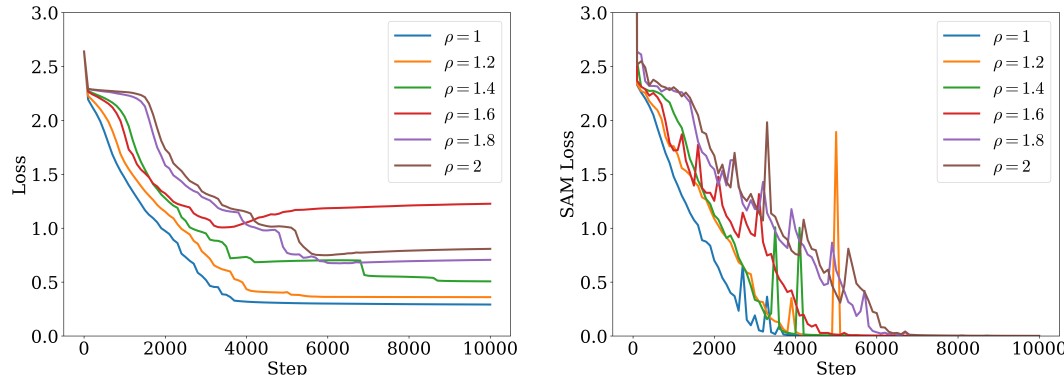

Figure 12: SAM on CIFAR-10 with ResNet-18: loss trajectories for different perturbation radii $\rho$.

Table 1: SAM on CIFAR-10 with ResNet-18: final-step metrics across perturbation radii $\rho$.

| $\rho$ | $f(x)$ | $f^{\text{SAM}}(x)$ | $\|\nabla f(x)\|$ | $\|\nabla f(x^+)\|$ |
|---|---|---|---|---|
| 1.0 | 0.2921 | 0.0009 | 6.0244 | 0.0169 |
| 1.2 | 0.3606 | 0.0009 | 7.1878 | 0.0091 |
| 1.4 | 0.5071 | 0.0014 | 8.9210 | 0.0089 |
| 1.6 | 1.2270 | 0.0011 | 2.8085 | 0.0105 |
| 1.8 | 0.7066 | 0.0016 | 3.1345 | 0.0303 |
| 2.0 | 0.8087 | 0.0017 | 2.4777 | 0.0245 |

# E  HALLUCINATED MINIMIZERS IN A LARGER DEEP NETWORK

We provide an additional SAM experiment on CIFAR-10 with ResNet-18 to demonstrate that hallucinated minimizers can arise in practical deep learning settings. We sweep the perturbation radius $\rho$ and track the training loss $f(x)$ together with the SAM loss $f^{\text{SAM}}(x)$. At the final step, we report the gradient norm $\|\nabla f(x)\|$ and the perturbed gradient norm $\|\nabla f(x^+)\|$.

We train ResNet-18 from scratch on CIFAR-10 using SAM with full-batch gradients. SGD is used as the base optimizer with learning rate $10^{-3}$ and momentum 0.9. We apply standard per-channel normalization and do not use data augmentation. We run 10,000 SAM steps and sweep $\rho \in \{1.0, 1.2, 1.4, 1.6, 1.8, 2.0\}$.

Figure 12 shows the trajectories of the training loss $f(x)$ and the SAM loss $f^{\text{SAM}}(x)$ over steps. For each $\rho$, the training loss decreases and stabilizes at a nonzero value. The stabilized value differs across $\rho$. In contrast, the SAM loss converges to a value close to zero for all $\rho$. This indicates a severe mismatch between $f(x)$ and $f^{\text{SAM}}(x)$: since $f^{\text{SAM}}(x)$ evaluates the loss at a local perturbation of $x$, it is typically expected to be no smaller than $f(x)$.

Table 1 summarizes the final-step metrics for each $\rho$. We report the final values of $f(x)$ and $f^{\text{SAM}}(x)$, along with the corresponding gradient norms. Consistent with Figure 12, $f^{\text{SAM}}(x)$ is near zero for all $\rho$, whereas $f(x)$ varies substantially across $\rho$. Moreover, $\|\nabla f(x)\|$ remains relatively large, while $\|\nabla f(x^+)\|$ is much smaller. These observations suggest that the SAM trajectory converges, since the per-step update magnitude in SAM is proportional to the perturbed gradient.

Taken together, Figure 12 and Table 1 show that the SAM trajectory converges while the training loss $f(x)$ remains strictly positive, whereas the SAM loss $f^{\text{SAM}}(x)$ approaches a value close to zero. This suggests that hallucinated minimizers can also occur in a larger network such as ResNet-18.

