# OpenReview forum: "Sharpness-Aware Minimization Can Hallucinate Minimizers"
_ICLR.cc/2026/Conference — Submitted to ICLR 2026_

### Official Review · Reviewer_Lhm8 · 2025-10-31

**Soundness:** 2
**Presentation:** 2
**Contribution:** 2
**Rating:** 2
**Confidence:** 3

**Summary:**

This paper focuses on a critical failure mode of Sharpness-Aware Minimization — hallucinated minimizers. Through theoretical analysis and experimental validation, it reveals the convergence defect of SAM in non-convex scenarios and uses a simple switching strategy to address this issue.

**Strengths:**

1. This paper completes strict proofs in non-convex and high-dimensional scenarios, covering isolated/non-isolated local maximizer sets.

2. It verifies the effectiveness of the switching strategy by sufficient experiments.

**Weaknesses:**

1. The primary advantage of SAM lies in its generalization ability rather than convergence. Thus, whether SAM converges to the true minimizers of the training loss is not a critical issue for understanding its behavior.

2. Even considering convergence, this issue should be addressed straightforwardly by decaying $\rho$ to zero during training.

3. The theoretical result is unsurprising, as the loss landscape is highly non-convex.

4. The switching strategy (SGD → SAM) is not novel; it has been extensively studied both theoretically and experimentally in prior works (e.g., Andriushchenko & Flammarion, ICML 2022; Zhou et al., ICLR 2025).

**Questions:**

See Weaknesses.

---

> ### Author Response · Authors · 2025-11-17
> **Authors' Response (1/2)**
>
> Dear Reviewer Lhm8,
>
> Thank you very much for your careful reading of our paper and for your detailed comments. We respond to your points one by one below.
>
> ---
>
> **Comment 1**
> "The primary advantage of SAM lies in its generalization ability rather than convergence. Thus, whether SAM converges to the true minimizers of the training loss is not a critical issue for understanding its behavior."
>
> **Response 1**
>
> We fully agree with the reviewer that the main strength of SAM lies in its generalization performance. However, the point we intend to emphasize in this paper is that, even before discussing generalization, there is a potential failure mode at the level of optimization itself: SAM can converge to a completely wrong point.
>
> SAM is originally designed to find flat minimizers. In the case of hallucinated minimizers that we analyze, the issue is not about “finding a better minimizer” or not, but rather about the algorithm failing to approach any true minimizer of the original loss and instead converging to an incorrect attractor. In such a situation, even if the generalization performance measured at that point does not look obviously poor, it should be interpreted as being obtained along a trajectory that is fundamentally different from the intended optimization path.
>
> In short, the core message of our work is that SAM may fail not only to find a flat minimizer, but in some cases may not even find a minimizer of the original loss at all.
>
> From this perspective, our analysis does not contradict prior work that focuses on the generalization benefits of SAM. Instead, it complements that line of research by highlighting a hidden risk at the level of optimization dynamics that can arise when SAM is used inappropriately.
>
> ---
>
> **Comment 2**
> "Even considering convergence, this issue should be addressed straightforwardly by decaying ρ to zero during training."
>
> **Response 2**
>
> Our theoretical and empirical results indeed indicate that hallucinated minimizers become less relevant when $\rho$ is sufficiently small. In that sense, we agree that reducing $\rho$ can alleviate the problem. However, we do not think that “decaying $\rho$ to zero” is a straightforward practical solution, for several reasons.
>
> - **Tuning $\rho$ inevitably involves trying large values.**
>    In practice, $\rho$ is almost always tuned by grid search. During this process, it is natural and common to try relatively large values of $\rho$, and our results show that precisely in this large-$\rho$ regime, convergence to hallucinated minimizers is likely. Because each SAM step requires two forward–backward passes, such failed runs are more costly than in standard GD.
>
> - **There is no simple, widely adopted rule for scheduling $\rho$.**
>    To our knowledge, there is no standard, simple SAM variant that adaptively adjusts $\rho$ during training in a reliable way while avoiding hallucinated minimizers. In practice, users still need to manually try different $\rho$ values or schedules, and some of these choices will inevitably fall into regimes where hallucinated minimizers appear.
>
> - **Sending $\rho$ to zero undermines SAM’s original goal.**
>    If $\rho$ is gradually reduced to (almost) zero, the method becomes increasingly similar to plain GD, and the specific benefit of SAM in preferring flatter minima is largely lost. In other words, $\rho$ encodes a trade-off between generalization and stability, and simply driving it to zero is not always desirable.
>
> For these reasons, our paper proposes a different approach: keep $\rho$ relatively large (to preserve the potential generalization benefits), but use GD in the early high-loss phase and switch to SAM only after the loss has decreased. This switching strategy reduces the probability of entering the region of attraction of hallucinated minimizers, while still allowing the use of larger $\rho$ than would be safe for SAM alone.

---

> ### Author Response · Authors · 2025-11-17
> **Authors' Response (2/2)**
>
> **Comment 3**
> "The theoretical result is unsurprising, as the loss landscape is highly non-convex."
>
> **Response 3**
>
> We agree that modern deep learning loss landscapes are highly non-convex, and that the presence of many local minima and saddle points is by now well understood. However, hallucinated minimizers are qualitatively different from these familiar critical points:
>
> - **Different from usual critical points.**
>   A hallucinated minimizer is neither a minimizer nor a saddle point of the original loss. It can lie close to a local maximizer of the original function and still be a stationary point of the SAM surrogate. In this sense, it is a critical point that exists only at the level of the SAM objective and not at the level of the underlying loss.
>
> - **Not a generic consequence of non-convexity.**
>   As we show in Appendix A.1, in the convex case hallucinated minimizers never appear. Thus, they are not an automatic byproduct of “non-convexity in general,” but a phenomenon that requires a specific interaction between minimizers and maximizers in sufficiently non-convex landscapes. This mechanism is described more explicitly in Theorem 2.1, which formalizes how such configurations give rise to hallucinated minimizers.
>
> - **Extension beyond 1D / quadratic analyses.**
>   Prior work has explored ideas related to misalignment between stationary points of the SAM surrogate and those of the original loss [1, 2], but typically in very restricted settings such as one-dimensional problems or quadratic functions. Our work takes this line of reasoning further by extending the analysis to general real-analytic, high-dimensional non-convex functions.
>
> From this perspective, our result is not just a restatement that “non-convex landscapes can behave strangely,” but a precise and rigorous description of a specific failure mode: how, and under what structural conditions, SAM can converge to points that are stationary for the surrogate but not for the original loss.
>
> [1] D. Si and C. Yun. Practical sharpness-aware minimization cannot converge all the way to optima. Neural Information Processing Systems, 2024.
> [2] P. L. Bartlett, P. M. Long, and O. Bousquet. The dynamics of sharpness-aware minimization: Bouncing across ravines and drifting towards wide minima. Journal of Machine Learning Research, 24(316):1–36, 2023.
>
> ---
>
> **Comment 4**
> "The switching strategy (SGD → SAM) is not novel; it has been extensively studied both theoretically and experimentally in prior works (e.g., Andriushchenko & Flammarion, ICML 2022; Zhou et al., ICLR 2025)."
>
> **Response 4**
>
> We agree that the SGD → SAM switching strategy itself is not new, and we explicitly cite prior works such as Zhou et al. (2025), and discuss the differences on p.9.
>
> Existing works mainly emphasize that applying SAM only in the later phase of training can preserve its generalization performance. In contrast, our focus is on the observation that using GD in the early phase makes it much easier to avoid convergence to hallucinated minimizers.
>
> In this sense, our contribution is to highlight an additional benefit of the switching strategy: it helps avoid this specific failure mode and, at the same time, empirically enlarges the range of perturbation radii that can be used safely.

---

### Official Review · Reviewer_KcAZ · 2025-10-31

**Soundness:** 2
**Presentation:** 2
**Contribution:** 2
**Rating:** 2
**Confidence:** 4

**Summary:**

This paper shows that SAM may converge to hallucinated minimizers, which can affect convergence and degrade performance. The authors prove the existence of such hallucinated minimizers and demonstrate that they can act as attractors. They conduct experiments to support their claims and suggest avoiding these hallucinated minimizers by switching from SGD to SAM.

**Strengths:**

This paper presents a novel perspective by identifying the existence of points where the gradient of the SAM objective vanishes, yet which are not stationary points of the original objective. The authors refer to these as hallucinated minimizers. Such points may be far from true minima and even close to maxima, thereby degrading performance. The paper provides a theoretical proof of their existence and discusses the conditions under which they can become attractors. Inspired by this theoretical insight, the authors suggest switching from SGD to SAM to address this issue, and the experimental results indicate that this can effectively alleviate SAM’s sensitivity to ρ. Overall, the exposition is reasonable and may contribute to a deeper understanding of SAM.

**Weaknesses:**

There exist points in neural network functions that are stationary for the SAM objective but not for the original objective, which seems rather intuitive. Although the authors make some effort to formalize the notion of hallucinated minimizers, from a technical standpoint I do not see a particularly significant contribution. Merely proving their existence does not provide much insight. Another main result of the paper is to show that hallucinated minimizers can become attractors. However, this requires quite strong regularity conditions and only demonstrates that the algorithm may converge to hallucinated minimizers; whether and when such convergence actually occurs in general case remain unclear.

The authors conduct several experiments to support their claims, but I find them unconvincing. Specifically, they argue that convergence to hallucinated minimizers is common for SAM in deep learning, yet I only observe divergence when the parameter ρ is relatively large. Unlike the toy examples, the authors do not report gradient norms or visualize the loss landscape, making it difficult to attribute the convergence failure to hallucinated minimizers. Moreover, I have not seen reports of SAM failing to converge in the literature, provided that ρ is chosen appropriately in practice.

The authors propose switching from SGD to SAM to address convergence failures, but this argument does not seem entirely coherent. They claim that hallucinated minimizers often occur near local maxima, and hence
are more likely to occur in high-loss regions.; however, I do not understand why a higher loss necessarily implies proximity to a local maximum. This seems to depend on local curvature rather than loss magnitude, suggesting that the authors may have conflated these concepts.

**Questions:**

I did not check the proofs line by line, but the authors state that “Theorem 2.1, whose proof requires the perturbation radius ρ to exceed the distance between a minimizer and a maximizer.” It should be noted that in high-dimensional spaces, distances typically grow with dimensionality. Does this imply that the theorem would require a larger ρ in high-dimensional settings, and thus is less likely to hold in deep learning scenarios?

In the experiments, the authors show that switching from SGD to SAM improves performance. Why does this differ from Figure 8 in Andriushchenko & Flammarion (2022), where they found that such a switch had an insignificant or even negative effect? Have the authors tried performing such a switch at different percentages?

---

> ### Author Response · Authors · 2025-11-17
> **Authors' Response (1/3)**
>
> Dear Reviewer KcAZ,
>
> Thank you very much for your careful reading of our paper and for your detailed comments. We respond to your points one by one below.
>
> **Comment 1**
> “There exist points in neural network functions that are stationary for the SAM objective but not for the original objective, which seems rather intuitive. Although the authors make some effort to formalize the notion of hallucinated minimizers, from a technical standpoint I do not see a particularly significant contribution. Merely proving their existence does not provide much insight.”
>
> **Response 1**
>
> As you point out, the fact that a stationary point of the SAM objective need not be a stationary point of the original loss is, by itself, somewhat intuitive. Our contribution is not just to note this possibility, but to give a precise theoretical description of when and how such points appear.
>
> - First, when the function is convex, hallucinated minimizers do not appear at all; this is shown in Appendix A.1. Thus, the phenomenon we study is not something that “automatically happens for any function,” but a feature that is specific to nonconvex structures.
> - In contrast, for sufficiently nonconvex real-analytic functions that have both minimizers and maximizers, our theorems guarantee the existence of hallucinated minimizers. There is no explicit restriction on the dimension, so the results apply directly in high-dimensional settings.
> - Related works in a similar spirit mostly analyze one-dimensional or quadratic functions [1, 2]. Our paper removes these restrictions and treats general (real-analytic) high-dimensional nonconvex functions, establishing the existence and structure of hallucinated minimizers and showing how they act as attractors for the SAM dynamics.
>
> In summary, the theoretical contribution of our work is to go beyond the informal statement that “such points might exist,” and to show that in general high-dimensional nonconvex settings (including neural networks) hallucinated minimizers can systematically exist and can act as attractors under SAM.
>
> [1] D. Si and C. Yun. Practical sharpness-aware minimization cannot converge all the way to optima. Neural Information Processing Systems, 2024.
> [2] P. L. Bartlett, P. M. Long, and O. Bousquet. The dynamics of sharpness-aware minimization: Bouncing across ravines and drifting towards wide minima. Journal of Machine Learning Research, 24(316):1–36, 2023.
>
> ---
>
> **Comment 2**
> “Another main result of the paper is to show that hallucinated minimizers can become attractors. However, this requires quite strong regularity conditions and only demonstrates that the algorithm may converge to hallucinated minimizers; whether and when such convergence actually occurs in general case remain unclear.”
>
> **Response 2**
>
> Thank you for pointing out this important issue. The convergence conditions in Theorem 3.2, especially the eigenvalue condition, are indeed not guaranteed to hold in full generality, and they are also difficult to verify explicitly in high-dimensional problems. We agree that this is a limitation of our current theoretical result.
>
> However, in the two-dimensional example of Figure 1, these conditions are satisfied exactly, and in this setting we observe that SAM does converge to the corresponding hallucinated minimizer. In addition, in our neural network experiments we can clearly observe cases where the SAM trajectory converges to points that match the behavior of hallucinated minimizers.
>
> In summary, we do not claim that SAM always converges to hallucinated minimizers. Our theorem shows that such convergence can occur when the stated conditions hold, and our experiments indicate that this type of convergence appears quite frequently in practice. Even when full convergence is not guaranteed, the presence of hallucinated minimizers can still distort the training trajectory and interfere with optimization, so we view them as a meaningful failure mode.

---

> ### Author Response · Authors · 2025-11-17
> **Authors' Response (2/3)**
>
> **Comment 3**
> “The authors conduct several experiments to support their claims, but I find them unconvincing. Specifically, they argue that convergence to hallucinated minimizers is common for SAM in deep learning, yet I only observe divergence when the parameter ρ is relatively large. Unlike the toy examples, the authors do not report gradient norms or visualize the loss landscape, making it difficult to attribute the convergence failure to hallucinated minimizers. Moreover, I have not seen reports of SAM failing to converge in the literature, provided that ρ is chosen appropriately in practice.”
>
> **Response 3**
>
> We address the reviewer’s concerns by clarifying four points.
>
> 1. **Convergence at high loss rather than divergence**
>
>    In our deep learning experiments with large ρ, the behavior we observe is not divergence in the sense of the loss or gradient blowing up. Instead, the trajectory moves to a high-loss region relatively quickly and then stays there without significant further change.
>    For example, in Figure 5, when ρ is large, the training loss increases to a certain level and then remains almost constant, which is different from classical divergence.
>
> 2. **Quantitative evidence from gradient norms**
>
>    Due to space limitations, we did not include all quantitative statistics in the paper, but we have a table that summarizes the mean loss and gradient norms over 80 runs for SAM and GD→SAM at different values of ρ:
>
>    | ρ   | Method   | f(x)   | ‖∇f(x)‖ | ‖∇f(x⁺)‖ |
>    |-----|----------|--------|---------|----------|
>    | 1.0 | SAM      | 0.0333 | 0.0053  | 0.0063   |
>    | 1.0 | GD→SAM   | 0.0141 | 0.0044  | 0.0053   |
>    | 1.5 | SAM      | 0.2603 | 0.0152  | 0.0555   |
>    | 1.5 | GD→SAM   | 0.0154 | 0.0046  | 0.0057   |
>    | 2.0 | SAM      | 2.2946 | 0.0630  | 0.0017   |
>    | 2.0 | GD→SAM   | 0.0185 | 0.0050  | 0.0061   |
>
>    For instance, when ρ = 2.0, SAM yields a very large loss, while the SAM-gradient norm becomes extremely small. This indicates convergence to a high-loss SAM stationary point rather than divergence.
>
> 3. **Loss landscape visualization beyond toy examples**
>
>    Figure 4 uses the same experimental setup as Figure 5, except that ρ is set to 1.8 and the total number of steps is increased by about a factor of 200. In this case, the SAM-gradient norm decreases to about 5 × 10⁻⁹, which is numerically small enough to be regarded as convergence.
>    Figure 4 visualizes the loss landscape around this final point and shows a location in the middle of a high-loss region where the SAM gradient is essentially zero. This matches the structure predicted by our theory for a hallucinated minimizer. Thus, these experiments on neural networks already go beyond low-dimensional toy examples.
>
> 4. **Relation to prior reports with “proper” ρ**
>
>    We agree that in the SAM literature, when a “proper” value of ρ is chosen, clear failure cases are rarely reported. However, in practice such a value is almost always found via grid search, and trying larger values of ρ during this search is both natural and common. Our results are precisely about what happens in this large-ρ regime.
>    Thus, there is no contradiction between prior work, which reports good behavior at well-tuned ρ, and our analysis, which focuses on failure modes that can occur for larger ρ values typically explored during tuning; rather, the two viewpoints are complementary.

---

> ### Author Response · Authors · 2025-11-17
> **Authors' Response (3/3)**
>
> **Comment 4**
> “The authors propose switching from SGD to SAM to address convergence failures, but this argument does not seem entirely coherent. They claim that hallucinated minimizers often occur near local maxima, and hence are more likely to occur in high-loss regions; however, I do not understand why a higher loss necessarily implies proximity to a local maximum. This seems to depend on local curvature rather than loss magnitude, suggesting that the authors may have conflated these concepts.”
>
> **Response 4**
>
> Our statement that hallucinated minimizers tend to appear in high-loss regions is intended as an intuitive interpretation rather than a fully rigorous theorem, and we agree that counterexamples can be constructed in principle.
>
> That said, our experiments (Figure 5) consistently show that, for large values of ρ, convergence to hallucinated minimizers is typically observed in high-loss regions. Moreover, when we apply the switching strategy and start SAM only after the loss has already decreased, we rarely observe such behavior, which suggests that it is indeed much less common in low-loss regions.
>
> In this sense, the “high-loss region” description should be understood as an empirical trend that summarizes what we observe in our experiments, not as a claim that loss magnitude alone determines proximity to a local maximizer. Local curvature is of course the more precise notion, and our use of loss level is meant as a practical and intuitive proxy for how these failure modes tend to appear in our setups.
>
> ---
>
> **Comment 5 (Question 1)**
> “I did not check the proofs line by line, but the authors state that ‘Theorem 2.1, whose proof requires the perturbation radius ρ to exceed the distance between a minimizer and a maximizer.’ It should be noted that in high-dimensional spaces, distances typically grow with dimensionality. Does this imply that the theorem would require a larger ρ in high-dimensional settings, and thus is less likely to hold in deep learning scenarios?”
>
> **Response 5**
>
> In deep learning settings, the range of ρ that can give rise to hallucinated minimizers can in fact become wider. In high-dimensional neural networks, it is well known that the set of minimizers often forms a manifold rather than a single isolated point, a phenomenon closely related to mode connectivity [3].
>
> Along paths connecting different minimizers on this manifold, there can be many points that behave like “maximizer-like” regions relative to a given minimizer. For each such minimizer–maximizer-like pair, Theorem 2.1 can potentially be applied. As the dimensionality grows, the number of such pairs naturally increases, and with it the opportunities for the conditions of Theorem 2.1 to be satisfied. This suggests that the admissible interval of ρ is not necessarily shrinking with dimension, and may even become richer.
>
> In this sense, the conditions for the existence of hallucinated minimizers can be more naturally satisfied in high-dimensional deep learning scenarios, rather than becoming less relevant.
>
> [3] P. Garipov, P. Izmailov, D. Podoprikhin, D. P. Vetrov, and A. G. Wilson. Loss surfaces, mode connectivity, and fast ensembling of DNNs. Advances in Neural Information Processing Systems (NeurIPS), 2018.
>
> ---
>
> **Comment 6 (Question 2)**
> “In the experiments, the authors show that switching from SGD to SAM improves performance. Why does this differ from Figure 8 in Andriushchenko & Flammarion (2022), where they found that such a switch had an insignificant or even negative effect? Have the authors tried performing such a switch at different percentages?”
>
> **Response 6**
>
> We have tried several switching points (different GD→SAM percentages), and the performance differences across these choices were generally very small, in line with the observations of Andriushchenko and Flammarion (2022). For this reason, we did not include all such variants in the main paper.
>
> The small performance gain we observe can be largely explained by the perturbation radius regime we consider. In SAM, larger values of ρ tend to promote flatter minima and can improve generalization, but they also increase the risk of converging to hallucinated minimizers. Our switching strategy reduces this risk, which in turn allows us to safely use relatively larger ρ values than plain SAM started from scratch.
>
> Figure 6 illustrates that the performance improvement from the SGD→SAM switch appears mainly in the large-ρ regime. This is consistent with our interpretation: the switch is most beneficial exactly where the risk of hallucinated minimizers is high but the potential generalization gains from a larger perturbation radius are also the greatest.

---

> > ### Comment · Reviewer_KcAZ · 2025-11-27
> >
> > Thank you for the authors’ response, which has addressed most of my concerns regarding the theoretical aspects. However, I still find the proposed switching strategy, including the authors’ replies to Comments 4 and 6, insufficiently convincing. I will raise my score to 4.

---

> > > ### Author Response · Authors · 2025-11-28
> > >
> > > Dear Reviewer KcAZ,
> > >
> > > Thank you for your additional comment and for raising your score.
> > >
> > > As you noted, we understand that our switching strategy (especially our response to Comment 4) is still not conveyed convincingly enough. In 80 repeated runs, we observed that when ρ is large, hallucinated minimizers are mainly observed in a high-loss regime. However, we acknowledge that our wording—“a large loss likely indicates proximity to a local maximizer”—was not rigorous and could be misleading.
> > >
> > > In the revision, we will revise that wording. Proposition A.1 shows that if local convexity holds near a minimizer, then hallucinated minimizers do not appear in that region. While these conditions do not always hold, we believe this can provide an intuitive explanation for our observation that hallucinated minimizers tend to appear less frequently in low-loss regimes.
> > >
> > > We appreciate your feedback, and we will further reduce any overstatement and make the logic clearer in the revision. If there are any additional specific points you would like us to address, we would be grateful to hear them.
> > >
> > > Sincerely,
> > > The Authors

---

### Official Review · Reviewer_46bm · 2025-10-31

**Soundness:** 3
**Presentation:** 4
**Contribution:** 3
**Rating:** 6
**Confidence:** 3

**Summary:**

This paper offers a highly original and rigorous analysis of SAM’s training dynamics near local maxima, a regime largely overlooked in prior work that has centered on saddle points. Under a clear and well-scoped setting, the authors prove that when local maxima exist, the algorithm can converge to hallucinated attractors; they precisely characterize the structure of these hallucinated local minima and establish their attractive dynamical properties. The proofs are cleanly organized, with the chosen mathematical tools applied appropriately. The exposition is exemplary—definitions and assumptions are explicit, intermediate lemmas are well-motivated, and the narrative flow makes technically demanding arguments easy to follow.

**Strengths:**

- The paper presents a theoretically complete and logically consistent treatment of the chosen problem, offering clear insights and rigorous analysis.
- Unlike prior studies that mainly focus on saddle points, this paper analyzes the failure mechanisms of SAM around local maxima, thereby broadening the scope of understanding.
- The authors establish the existence of hallucinated minima and provide a systematic characterization of their dynamics and the mechanisms that induce them
- The flow from definitions to assumptions, theorems, and lemmas is logically coherent, and the core ideas are clearly articulated, allowing readers to follow even complex arguments with ease.

**Weaknesses:**

- The significance of the problem itself is not convincingly justified, raising doubts about its broader relevance and practical impact.
- A key limitation of the theoretical analysis is its reliance on strong, idealized assumptions. The framework presumes the existence of isolated local maximizers and real-analytic loss functions, and requires a perturbation radius that aligns with the specific geometry of the landscape. These conditions are rarely met in the high-dimensional and non-analytic nature of modern neural networks trained with stochastic optimization.
- The most detailed analyses and visualizations are confined to a simple 2-layer MLP on MNIST. The ResNet-18/CIFAR-100 results, while suggestive, lack deeper probes of the loss landscape or properties of the converged points. Broader coverage across architectures, datasets, and realistic ρ ranges would strengthen the empirical case
- The “hallucination” behavior, though interesting, appears modest in commonly used SAM regimes and is not yet shown to materially affect performance across realistic workloads. Clearer quantification of effect sizes and frequency under standard training settings would help establish practical impact.
- Although the phenomenon is intriguing, its real-world impact remains under-substantiated; stronger evidence across architectures and tasks is needed to establish practical relevance.

**Questions:**

1. One advantage for using SAM is its robustness to label noise. Your paper demonstrates a significant failure mode of SAM in a clean-label setting. Does this failure mode—converging to hallucinated minimizers—persist under noisy label conditions, or is it mitigated in the very regime where SAM is considered most beneficial?
2. Your central hypothesis is that the SGD→SAM switching strategy is effective because SGD avoids high-loss regions near local maximizers, which are the source of hallucinated minimizers. However, the paper lacks direct empirical evidence that local maximizers are a significant feature of the initial loss landscape. How can you empirically verify the existence and prevalence of local maximizers near random initialization for your experimental setups

---

> ### Author Response · Authors · 2025-11-17
> **Authors' Response (1/3)**
>
> Dear Reviewer 46bm,
>
> Thank you very much for your thoughtful and detailed review. We appreciate your positive assessment of the theory and presentation, and we address your concerns point by point below.
>
> ---
>
> **Comment 1**
> *“The significance of the problem itself is not convincingly justified, raising doubts about its broader relevance and practical impact.”*
>
> **Response 1**
>
> The theoretical value of this work is that it shows SAM can lead not only to divergence or convergence to standard local minimizers or saddle points, but also to a previously unreported type of training failure. In our analysis, SAM may converge to hallucinated minimizers, which are not minimizers of the original loss.
>
> From a practical point of view, the key point is that when the perturbation radius is sufficiently large, the algorithm can converge to such hallucinated minimizers. At present, choosing an appropriate perturbation radius in practice is essentially done by grid search. This means that, in real experiments, users are quite likely to encounter these failure modes. For this reason, we believe that the phenomenon studied in this paper has concrete practical relevance, in addition to its theoretical interest.
>
> ---
>
> **Comment 2**
> *“A key limitation of the theoretical analysis is its reliance on strong, idealized assumptions. The framework presumes the existence of isolated local maximizers and real-analytic loss functions, and requires a perturbation radius that aligns with the specific geometry of the landscape. These conditions are rarely met in the high-dimensional and non-analytic nature of modern neural networks trained with stochastic optimization.”*
>
> **Response 2**
>
> We agree that the assumptions deserve careful discussion, but we believe they are not as idealized as they may first appear. We explain our viewpoint below.
>
> First, regarding the real-analytic assumption and isolated maximizers:
>
> - As stated in Theorem 2.2, the “isolated” property is not an additional independent assumption; it can be derived from the real-analytic assumption in our setting. Thus, the key question is how reasonable the real-analytic assumption itself is.
> - Neural networks without max-type operations (such as ReLU) typically define real-analytic functions, as we explain in detail in Appendix A.3. In this sense, the real-analytic assumption is not excessively unrealistic, but is compatible with a nontrivial class of practical architectures.
> - Moreover, recent theoretical works on the convergence of SAM also adopt similar assumptions [1, 2].
>
> Second, regarding the perturbation radius:
>
> - We do not assume a single finely tuned value of the perturbation radius. In Theorem 2.1, the admissible perturbation radii form an interval (a range of ρ values), rather than a single point.
> - In addition, by changing the combination of minimizers and maximizers in Theorem 2.1, the range of admissible ρ can be further extended.
>
> Finally, our theoretical results do not impose any explicit restriction on the dimension, so high-dimensional settings are included in our framework. This is consistent with our neural network experiments, where we directly observe both the existence of hallucinated minimizers and convergence to them.
>
> [1] P. Khanh, H.-C. Luong, B. Mordukhovich, and D. Tran. *Fundamental convergence analysis of sharpness-aware minimization.* Advances in Neural Information Processing Systems (NeurIPS), 2024.
>
> [2] D. Oikonomou and N. Loizou. *Sharpness-aware minimization: General analysis and improved rates.* International Conference on Learning Representations (ICLR), 2025.

---

> ### Author Response · Authors · 2025-11-17
> **Authors' Response (2/3)**
>
> **Comment 3**
> *“The most detailed analyses and visualizations are confined to a simple 2-layer MLP on MNIST. The ResNet-18/CIFAR-100 results, while suggestive, lack deeper probes of the loss landscape or properties of the converged points. Broader coverage across architectures, datasets, and realistic ρ ranges would strengthen the empirical case.”*
>
> **Response 3**
>
> We theoretically show that hallucinated minimizers can arise in deep learning settings, and we confirm this phenomenon empirically in a simple example (a 2-layer MLP on MNIST). Based on this combination of theory and experiments, we believe that the same mechanism also operates in more complex and larger-scale tasks. However, in such large tasks, precisely identifying whether a given point is in fact a hallucinated minimizer requires extremely high computational cost and is currently very challenging in practice. We agree that this would be valuable, and we view a more systematic investigation in such large-scale settings as an important direction for future work.
>
> We would also like to emphasize that “realistic” ranges of ρ are highly task-dependent and not fixed. At present, choosing an appropriate ρ essentially relies on grid search, and it is a hyperparameter that requires as much care as the learning rate. For this reason, in practical settings it is not easy for users to reliably avoid the region of attraction of hallucinated minimizers, which further underscores the practical relevance of our findings.
>
> ---
>
> **Comment 4**
> *“The ‘hallucination’ behavior, though interesting, appears modest in commonly used SAM regimes and is not yet shown to materially affect performance across realistic workloads. Clearer quantification of effect sizes and frequency under standard training settings would help establish practical impact.
> Although the phenomenon is intriguing, its real-world impact remains under-substantiated; stronger evidence across architectures and tasks is needed to establish practical relevance.”*
>
> **Response 4**
>
> As discussed above, we provide a detailed theoretical analysis of hallucinated minimizers, and the assumptions used there are formulated so that they can apply to general deep learning tasks. Our experiments focus on relatively small models, but their role is to serve as concrete examples illustrating the phenomenon, rather than to exhaust all possible architectures.
>
> Due to practical limitations in computation and resources, we were not able to perform the same level of analysis on much larger models. However, given our theory and the current experimental evidence, we believe that hallucinated minimizers are also likely to exist in more realistic large-scale settings.

---

> ### Author Response · Authors · 2025-11-17
> **Authors' Response (3/3)**
>
> **Comment 5 (Question 1)**
> *“One advantage for using SAM is its robustness to label noise. Your paper demonstrates a significant failure mode of SAM in a clean-label setting. Does this failure mode—converging to hallucinated minimizers—persist under noisy label conditions, or is it mitigated in the very regime where SAM is considered most beneficial?”*
>
> **Response 5**
>
> If we fix a noisy-label dataset and train on it repeatedly, our analysis and the phenomenon essentially do not change. A “hallucinated minimizer” is not directly about generalization performance; it is a failure mode of the optimization dynamics itself, where SAM converges to points that are not minimizers of the original loss. Therefore, with a fixed noisy dataset, the possibility of converging to hallucinated minimizers remains, just as in the clean-label setting.
>
> On the other hand, if the noisy labels change over time (for example, if each epoch sees a substantially different noisy labeling), then convergence to the same hallucinated minimizer may be mitigated to some extent, because the underlying loss landscape being optimized keeps changing.
>
> ---
>
> **Comment 6 (Question 2)**
> *“Your central hypothesis is that the SGD→SAM switching strategy is effective because SGD avoids high-loss regions near local maximizers, which are the source of hallucinated minimizers. However, the paper lacks direct empirical evidence that local maximizers are a significant feature of the initial loss landscape. How can you empirically verify the existence and prevalence of local maximizers near random initialization for your experimental setups?”*
>
> **Response 6**
>
> In our theoretical development, we assume the existence of local maximizers, but in practical neural networks it is difficult to guarantee the existence of perfectly well-defined local maximizers. Nevertheless, hallucinated minimizers can still arise in neural networks because what is actually needed is not a maximizer itself, but an appropriate superlevel set structure.
>
> More precisely, if there exists a point in a superlevel set whose distance to the original minimizer is locally maximal, then, by our mechanism, one can still guarantee the existence of a corresponding hallucinated minimizer.

---

### Official Review · Reviewer_E2P5 · 2025-11-03

**Soundness:** 3
**Presentation:** 3
**Contribution:** 3
**Rating:** 4
**Confidence:** 3

**Summary:**

The paper shows that Sharpness-Aware Minimization can converge to “hallucinated minimizers” — points that minimize the SAM surrogate but are not stationary, let alone minimizers, of the original loss.
The authors prove that such spurious attractors exist in very general non-convex, real-analytic settings, describe their manifold structure, and confirm their appearance in deep-net experiments with both full-batch and stochastic gradients.
To avoid them they propose a trivial “GD-first, SAM-later” switching rule that keeps large perturbation radii and improves both training stability and test accuracy.

**Strengths:**

1. The use of a constructed smooth non-convex function (Figure 1) effectively demonstrates how the SAM objective can create spurious minima not present in the original function, making the phenomenon intuitive.

2. Beyond identifying the problem, the authors propose a practical fix—using a damped version of the gradient—to prevent convergence to hallucinated minimizers, enhancing the reliability of SAM in real applications.

3. The paper provides a rigorous theoretical proof that SAM can converge to non-minimizing points (hallucinated minimizers), which challenges the common assumption that SAM always guides optimization toward meaningful solutions.

**Weaknesses:**

1. While the paper includes empirical results, they are primarily based on simple synthetic examples or small-scale models; more extensive validation on large-scale deep learning tasks would strengthen the practical relevance.

2. The proposed remedy is simple and effective in theory, but the paper does not thoroughly compare it against other SAM variants or alternative flatness-promoting methods, limiting the assessment of its relative advantages.

3. In the last part of this paper, authors choose s to use plain gradient descent for the first 10% of training iterations. There is a lack of quantitative metrics to determine when to switch to the SAM method.

4. In Theorem 3.2, the authors claim that a poor initialization may lead to hallucinated minimizers. However, there is no theoretical analysis indicating that plain gradient descent is more likely to yield a good initialization than SAM.

**Questions:**

1. If the  perturbation radius pho is small enough, will the hallucinated minimizers vanish?

---

> ### Author Response · Authors · 2025-11-16
> **Authors' Response (1/3)**
>
> Dear Reviewer E2P5,
>
> Thank you for your careful review and constructive comments. Below we respond to your points one by one.
>
> ---
>
> **Comment 1**
> *“While the paper includes empirical results, they are primarily based on simple synthetic examples or small-scale models; more extensive validation on large-scale deep learning tasks would strengthen the practical relevance.”*
>
> **Response 1**
>
> We agree that additional experiments on large-scale deep learning tasks would strengthen the practical impact of the paper. Here we explain why we did not include such large-scale experiments.
>
> - The main contribution of this work is theoretical: we prove in a real-analytic non-convex setting that SAM can converge to hallucinated minimizers. The neural network experiments play the role of proof-of-concept, confirming that the phenomenon predicted by the theory also appears in actual models. Taken together, the theory and medium-scale experiments suggest that hallucinated minimizers may also arise in large-scale deep learning tasks.
> - It is also very difficult in practice to verify whether a *specific* point in a large-scale model is a hallucinated minimizer. For small models, in a deterministic setting, we can train long enough and locate hallucinated minimizers with reasonable accuracy. Applying the same level of verification to large models would require prohibitively high computational cost.
>
> Therefore, in this paper we restrict our scope to theoretical analysis and medium-scale experiments, and we leave a systematic empirical study in larger-scale settings as important future work.
>
> ---
>
> **Comment 2**
> *“The proposed remedy is simple and effective in theory, but the paper does not thoroughly compare it against other SAM variants or alternative flatness-promoting methods, limiting the assessment of its relative advantages.”*
>
> **Response 2**
>
> We agree with the reviewer that it would be useful to compare our method with a wider range of SAM variants. Here we explain why we focus on the standard SAM formulation in this paper, and why we expect the behavior of hallucinated minimizers to depend on the specific variant.
>
> We also expect that the behavior of hallucinated minimizers can depend on the specific variant.
>
> - For variants where the SAM gradient is mainly determined by the current position in a deterministic way (e.g., ASAM [1]), our mechanism is likely to remain relevant, so hallucinated minimizers may still appear.
> - In contrast, for variants where the ascent direction has strong stochastic components or depends heavily on history (e.g., FriendlySAM [2], VaSSO [3]), the same effect may be weaker, or it may manifest in a different way.
>
> Because of this, it is difficult to analyze many SAM variants in a precise way, both theoretically and experimentally, within a single paper. In this work, we therefore concentrate on the standard SAM formulation and study hallucinated minimizers for this baseline in detail, both in theory and in experiments. For the remedy, we also intentionally propose the simplest possible modification on top of standard SAM, so that the change to the original algorithm is minimal.
>
> For these reasons, in the present paper we restrict our attention to standard SAM, and leave a detailed analysis of SAM variants as interesting future work.
>
> [1] J. Kwon, J. Kim, H. Park, and I. K. Choi. *ASAM: Adaptive sharpness-aware minimization for scale-invariant learning of deep neural networks.* International Conference on Machine Learning (ICML), 2021.
>
> [2] T. Li, P. Zhou, Z. He, X. Cheng, and X. Huang. *Friendly sharpness-aware minimization.* IEEE/CVF Conference on Computer Vision and Pattern Recognition (CVPR), 2024.
>
> [3] B. Li and G. Giannakis. *Enhancing sharpness-aware optimization through variance suppression.* Advances in Neural Information Processing Systems (NeurIPS), 2024.

---

> ### Author Response · Authors · 2025-11-17
> **Authors' Response (2/3)**
>
> **Comment 3**
> *“In the last part of this paper, authors choose to use plain gradient descent for the first 10% of training iterations. There is a lack of quantitative metrics to determine when to switch to the SAM method.”*
>
> **Response 3**
>
> The reviewer points out that, in our experiments, we use plain GD for the first 10% of iterations and then switch to SAM. However, in practice this ratio does not seem to be particularly critical. We explain our choice below.
>
> - First, prior work already provides some guidance on how important the GD:SAM ratio is for final performance.
>   Andriushchenko and Flammarion [4] report that using GD for 0–50% of the steps yields almost the same performance, and Zhou et al. [5] show that using GD for the first 90% and SAM only for the last 10% can still achieve good performance. In our additional experiments (not included in the paper), changing the GD fraction between 10–50% also led to almost no change in test accuracy. Since these observations are consistent with existing results, we did not include them in the main text.
>
> - Second, the main focus of our paper is not performance tuning, but avoiding convergence to hallucinated minimizers when the perturbation radius is large. From this perspective, what matters is not the late phase of training, but how much of the early high-loss phase is run with GD.
>   Empirically, when the GD phase was too short (about 5% or less of all steps), we still observed cases where the trajectory fell into a hallucinated minimizer. When we used GD for about 10% or more of the steps, the iterate had already reached a low-loss region by the time SAM started, and convergence to hallucinated minimizers disappeared in a stable way. We therefore set the GD fraction to 10% to show that even a small amount of GD is sufficient to avoid hallucinated minimizers. Increasing the GD fraction does not add any burden in terms of avoiding this failure mode, and, as noted above, has almost no effect on final performance.
>
> In summary, the exact GD fraction is not very sensitive in practice, and we fixed it to 10% because this small fraction already has the desired effect. In practical applications, using a GD fraction in the range of 20–50% would be a safe choice.
>
> [4] M. Andriushchenko and N. Flammarion. *Towards understanding sharpness-aware minimization.* International Conference on Machine Learning (ICML), 2022.
>
> [5] Z. Zhou, M. Wang, Y. Mao, B. Li, and J. Yan. *Sharpness-aware minimization efficiently selects flatter minima late in training.* International Conference on Learning Representations (ICLR), 2025.
>
> ---
>
> **Comment 4**
> *“In Theorem 3.2, the authors claim that a poor initialization may lead to hallucinated minimizers. However, there is no theoretical analysis indicating that plain gradient descent is more likely to yield a good initialization than SAM.”*
>
> **Response 4**
>
> We believe there may be a small misunderstanding about what we mean by “initialization” in Theorem 3.2, so we clarify it here.
>
> In Theorem 3.2, initialization simply refers to the parameter point at which we start applying SAM. The key issue is how close this point is to a hallucinated minimizer and its region of attraction, not which algorithm produced it.
>
> In Theorem 2.1, we also explain that the hallucinated minimizer is located near a maximizer of the original loss, that is, mainly in a high-loss region. Therefore:
>
> - If SAM is started directly from a high-loss region near such a maximizer, the trajectory is more likely to be attracted to a hallucinated minimizer.
> - If SAM is started from a low-loss region, this risk is reduced.
>
> This behavior is also observed empirically in Figure 5. That figure shows that whether SAM converges to a hallucinated minimizer depends on the loss level at which SAM is started. Combining the theory and experiments, we see that starting SAM from a lower-loss point helps avoid hallucinated minimizers.
>
> In contrast to SAM, plain GD tends to reduce the loss in a stable way in the early phase. For this reason, we first use GD to move the parameters into a low-loss region, and then switch to SAM. In this context, our use of the term “initialization” refers to starting SAM from a low-loss state, rather than the usual notion of weight initialization in deep learning.

---

> ### Author Response · Authors · 2025-11-17
> **Authors' Response (3/3)**
>
> **Comment 5 (Question 1)**
> *“If the perturbation radius ρ is small enough, will the hallucinated minimizers vanish?”*
>
> **Response 5**
>
> As the question suggests, in typical situations hallucinated minimizers tend to disappear when the perturbation radius ρ becomes sufficiently small.
>
> - First, even in the construction used in Theorem 1, ρ has to be larger than the distance between a maximizer and a minimizer of the original function for the theorem to apply. If ρ is smaller than this distance, that particular construction no longer works.
> - Intuitively, near a minimizer the gradient usually points back toward the minimizer, so when ρ is very small, it is difficult for SAM to create a new minimizer outside the original one.
> - In particular, when the neighborhood around the minimizer is locally convex, small ρ does not generate hallucinated minimizers; this is shown in Proposition A.1.
>
> On the other hand, in nonconvex settings it is possible to construct pathological counterexamples. For example, in a two-dimensional function, one can design a gradient flow that keeps circling around a minimizer and adjust the gradient so that, at certain angles, it points slightly outward. In such an artificially constructed case, it is theoretically possible to obtain a hallucinated minimizer even for small ρ.
>
> However, these examples are highly contrived. In our neural network experiments, we did not observe such behavior when ρ was sufficiently small. At the same time, it is difficult to predict in advance what “sufficiently small” ρ should be in a given problem, so in practice it remains challenging to guarantee that hallucinated minimizers are always avoided.

---

### Comment · Area_Chair_GgDn · 2025-11-27
**Request for Timely Response to Authors’ Rebuttal and Discussion**

Dear Reviewers,

I hope you are doing well. The authors have now submitted their rebuttal for the paper under your review. At this stage, your timely response is essential for ensuring a smooth discussion phase.

Could you please review the rebuttal at your earliest convenience and share your updated thoughts? If there are points that require further discussion among the reviewers, please feel free to initiate or join the conversation on the discussion thread.

Your prompt input will greatly help us maintain the review timeline. Thank you very much for your efforts and valuable contributions.

Best regards,

AC

---

### Author Response · Authors · 2025-12-03
**Update in the Revised Version: Additional CIFAR-10/ResNet-18 Results**

Dear Reviewers,

We have posted a revised version of our manuscript that incorporates the feedback from the review process.

In particular, to address the concern that our empirical validation relies primarily on synthetic examples and relatively small models and would benefit from additional evidence in a larger deep-learning setting (E2P5, 46bm), we conducted an additional experiment on CIFAR-10 with ResNet-18 and added the results to Appendix E.

In this experiment, we sweep the perturbation radius $\rho$ and report the trajectories of the training loss $f(x)$ and the SAM loss $f^{\mathrm{SAM}}(x)$ across training steps. To provide more direct quantitative evidence about the convergence behavior, we also report the gradient norm $|\nabla f(x)|$ and the perturbed gradient norm $|\nabla f(x^+)|$ at the final step. This also addresses the request for gradient-norm evidence raised by reviewer KcAZ.

We hope these additions strengthen the empirical support by showing that hallucinated minimizers can also be observed in a larger network setting.

Sincerely,
The Authors

---

### Meta-Review · Area_Chair_fPcN · 2026-01-07

**Summary:**

* Practical Significance and Scale: Multiple reviewers (46bm, E2P5) argued that the phenomenon was demonstrated primarily on synthetic data or small-scale models (MNIST/MLP). They questioned if these "hallucinated minimizers" actually occur or affect performance in modern, large-scale deep learning workloads.


* Novelty of the Solution: Reviewers noted that the proposed remedy (switching from SGD to SAM) is a known technique in existing literature. They felt the paper lacked quantitative metrics for when to switch, making the "fix" seem heuristic.


* Regularity Assumptions: Reviewers 46bm and KcAZ pointed out that the theoretical proofs rely on strong assumptions, such as real-analytic loss functions and isolated local maximizers, which may not hold in high-dimensional neural networks.

* Intuition vs. Impact: Reviewer KcAZ and Lhm8 argued that it is mathematically intuitive/expected for a surrogate objective (SAM) to have different stationary points than the original loss, questioning the depth of the contribution.

**Reviewer Concerns:**

Concerns Addressed

* Large-Scale Evidence: The authors conducted new experiments on CIFAR-10 with ResNet-18, reporting gradient norms ($||\nabla f(x)||$ and $||\nabla f(x^+)||$) to provide quantitative proof of convergence to hallucinated points in larger models

* Theoretical Clarity on "Initialization": The authors clarified that Theorem 3.2 refers to the starting point of SAM, not weight initialization. They explained how starting SAM from a low-loss region (via initial GD) avoids the attractors located near high-loss maximizers

* Dimension and Radius ($\rho$): The authors effectively argued that high-dimensionality actually increases the likelihood of "maximizer-like" regions, potentially making the existence of hallucinated minimizers more common in deep learning than in toy examples

Outstanding Concerns

* Switching Heuristics: The authors admitted that the 10% switching point is not strictly optimized but based on empirical stability. The lack of a "principled" metric for the switch remains a weakness.

* Practical Performance Impact: While the authors showed convergence to "wrong" points, they did not definitively prove that these points lead to significantly worse test performance compared to standard SAM across many diverse architectures

* Assumption: While the authors defended this for networks without ReLU, the gap between the theoretical requirement and modern non-smooth architectures (ReLU, MaxPool) remains

* The concern of the conclusion being expected remains.

**Reviewer Scores:**

* E2P5	Mixed. The authors somewhat addressed the "scale" concern with ResNet-18 results and clarified the switching logic.

* 46bm	The reviewer praised the theory's originality but remained skeptical of "real-world impact". The CIFAR results help, but may not fully bridge the "practical relevance" gap for this reviewer.


* KcAZ	This reviewer already raised their score to 4 after the rebuttal, noting that theoretical concerns were addressed but the switching strategy was still "insufficiently convincing".


* Lhm8	This reviewer felt the problem itself was "not a critical issue" because SAM is for generalization, not convergence. The authors' philosophical defense  is unlikely to change such a fundamental disagreement on the paper's premise.

---

### Decision · Program_Chairs · 2026-01-26

Reject